

# Global peatlands under future climate - seamless model projections from the Last Glacial Maximum

Jurek Müller[1,2] and Fortunat Joos[1,2]

[1]Climate and Environmental Physics, Physics Institute, University of Bern, Bern, Switzerland
[2]Oeschger Centre for Climate Change Research, University of Bern, Bern, Switzerland

**Correspondence:** Jurek Müller (jurek.mueller@climate.unibe.ch)

**Abstract.**

Peatlands are diverse wetland ecosystems distributed mostly over the northern latitudes and tropics. Globally they store a large portion of the global soil organic carbon and provide important ecosystem services. The future of these systems under continued anthropogenic warming and direct human disturbance has potentially large impacts on atmospheric $CO_2$ and climate. We performed global long term projections of peatland area and carbon over the next 5000 years using a dynamic global vegetation model forced with climate anomalies from ten models of the Coupled Model Intercomparison Project (CMIP6) and three scenarios. These projections are continued from a transient simulation from the Last Glacial Maximum to the present to account for the full transient history. Our results suggest short to long term net losses of global peatland area and carbon, with higher losses under higher emission scenarios. Large parts of today's active northern peatlands are at risk. Conditions for peatlands in the tropics and, in case of mitigation, eastern Asia and western north America improve. Factorial simulations reveal committed historical changes and future rising temperature as the main driver of future peatland loss and increasing precipitations as driver for regional peatland expansion. Additional simulations forced with two CMIP6 scenarios extended transiently beyond 2100, show qualitatively similar results to the standard scenarios, but highlight the importance of extended future scenarios for long term carbon cycle projections. The spread between simulations forced with different climate model anomalies suggests a large uncertainty in projected peatland variables due to uncertain climate forcing. Our study highlights the importance of quantifying the future peatland feedback to the climate system and its inclusion into future earth system model projections.

## 1 Introduction

Peatlands are a wetland type that is characterized by thick layers of accumulated organic carbon facilitated by permanently waterlogged conditions (Moore, 1989; Blodau, 2002). Suitable conditions can vary globally and can depend on local hydrology, topography, climate, and vegetation (Gorham, 1957) resulting in multiple forms from minerotrophic fens to ombrotrophic bogs





and forested tropical peat swamps (Rydin and Jeglum, 2013; Page and Baird, 2016; Lindsay, 2018). Although global peatlands cover only 3 % of the global land area (Xu et al., 2018b), they have an integral role in the global carbon cycle (Gorham, 1991; Yu, 2011; Page et al., 2011). They function as long term carbon stores holding up to a third of the total global soil organic carbon (Page et al., 2011; Yu, 2012). Most of today's peatlands, formed and accumulated carbon over the last 12,000 years,

driven by deglacial climate change and ice sheet retreat (e.g. Halsey et al., 2000; Gajewski et al., 2001; MacDonald et al., 2006; Gorham et al., 2007; Yu et al., 2010; Ruppel et al., 2013; Morris et al., 2018; Treat et al., 2019; Müller and Joos, 2020). Peatlands often are at the same time long term sinks of carbon (e.g. Gorham et al., 2012; Lähteenoja et al., 2012; Leifeld et al., 2019) as well as large natural sources of methane (e.g. Frolking and Roulet, 2007; LAI, 2009; Korhola et al., 2010; Yu et al., 2013; Packalen et al., 2014; Dommain et al., 2018). The net radiative effect over the Holocene has been a cooling (Frolking

and Roulet, 2007).

Apart from their function as long term carbon stores, and net carbon sinks, peatlands provide many more important ecosystem services (Kimmel and Mander, 2010; Page and Baird, 2016). Peatlands act as hydrological buffers providing purified drinking water (Xu et al., 2018a). As unique ecosystems peatlands are a habitat to many rare and specialized species and thus preserve global biodiversity (Minayeva and Sirin, 2012). Culturally they can serve recreational and spiritual functions. For

environmental researchers they provide a unique archive for environmental and cultural change over millennia (de Jong et al., 2010).

Direct and indirect anthropogenic disturbances however have exerted increasing pressures on global peatlands, threatening their important ecosystem services and potentially putting large carbon stocks at risk (Posa et al., 2011; Goldstein et al., 2020). Direct disturbances include peatland drainage for land-use conversion and peat mining, which has lead to large carbon

losses in temperate and tropical regions (Hergoualc'h and Verchot, 2011; Dohong et al., 2017; Leifeld et al., 2019; Dommain et al., 2018; Hoyt et al., 2020). Low water tables after drainage also facilitate increased peat burning (Turetsky et al., 2015; Page and Hooijer, 2016). Drainage of agricultural areas can also affect neighbouring unmanaged peatlands (Beauregard et al., 2020). Degradation following past land-use conversion will continue to release large amounts of carbon over decades to come (Leifeld and Menichetti, 2018). Given prompt action, this committed and additional carbon loss could be partly mitigated with

large-scale restoration and re-wetting efforts (Warren et al., 2017; Nugent et al., 2019; Günther et al., 2020) in conjunction with strong protection policies (Humpenöder et al., 2020; Wibisana and Setyorini, 2021).

Indirect human disturbances are mediated through anthropogenic climate change which is rapidly changing the boundary conditions for global peatlands. Mean annual precipitation is projected to increase in regions of large peatland extent such as the northern high latitudes and south east Asia (IPCC, 2013), possibly improving conditions for peatland development and carbon

accumulation. However, increases in precipitation are often offset by increased evapotranspiration under a warmer climate. Temperatures are projected to disproportionately increase in the northern high latitudes (IPCC, 2013), where the largest portion of global peatlands reside (Xu et al., 2018b). Industrial warming has already lead to increases in peatland evapotranspiration (Helbig et al., 2020b) leading to a widespread drying trend in the peatlands of northern Europe (Swindles et al., 2019; Zhang et al., 2020) and eastern Canada peatlands (Pellerin and Lavoie, 2003). The water table is an important regulator in peatland

ecosystems with complex feedbacks to vegetation and carbon cycling (Sawada et al., 2003; Zhong et al., 2020). A water table





draw down leads to increased fire frequency (Turetsky et al., 2015) and to a shift in vegetation cover from moss dominated to shrub and tree dominated (Pellerin and Lavoie, 2003; Talbot et al., 2010; Pinceloup et al., 2020; Beauregard et al., 2020). Lower water tables also lead to the exposure of progressively deeper peat layers to oxic conditions, increasing decomposition (Ise et al., 2008; Zhong et al., 2020). Higher temperatures also generally lead to higher decomposition rates with both increases

in measured $CO_2$ (Hopple et al., 2020; Kluber et al., 2020) and methane emissions (Turetsky et al., 2014). Although some studies suggest deep peat carbon to be robust under future warming (Wilson et al., 2016). In the northern high latitudes this might be offset by increases in plant productivity, even leading to net increases in carbon accumulation (Charman et al., 2013; Gallego-Sala et al., 2018).

About 46 % of northern peatlands are underlain by permafrost (Hugelius et al., 2020), which in some regions is quickly

thawing as a response to global warming (Camill, 2005; Lara et al., 2016; Mamet et al., 2017). Permafrost thaw is projected to accelerate dramatically depending on the future scenario (Lawrence et al., 2012; Guo and Wang, 2016). Permafrost peatlands have been found to often collapse after thaw and form thermokarst landscapes and collapse-scar wetlands (Payette et al., 2004; Olefeldt et al., 2016; Magnússon et al., 2020) characterized by carbon loss and high methane emissions (Jiang et al., 2020; Voigt et al., 2019; Turetsky et al., 2020; Estop-Aragonés et al., 2020). Given sustained inundation, renewed and invigorated

accumulation is assumed to set in after collapse, leading to an eventual return to a net cooling effect after decades to millennia of net warming (Swindles et al., 2015; Jones et al., 2017; Magnússon et al., 2020). However, some peatlands show an increase in carbon accumulation already directly after thaw (Estop-Aragonés et al., 2018).

Investigating the potential future trajectories of global peatlands is of great importance, given the multiple pressures on peatlands as unique ecosystems and carbon stores, which will further increase with future climate and land-use change. Although

the potential feedbacks between peatlands, the carbon cycle and the climate system could be immense, peatlands are in general still not included in state of the art Earth System Models (ESM) (Loisel and Bunsen, 2020), with only few exceptions (Schuldt et al., 2013). A large part of the global carbon cycle is thus also missing in the future climate and carbon cycle projections used for the determination of international climate mitigation targets, such as the sixth phase of the Climate Model Intercomparison Project (CMIP6) (Eyring et al., 2016). Different approaches have been used to independently project different aspects of future

peatland dynamics under future scenarios. Paleo data driven approaches can be used to investigate future peatland carbon accumulation rates (Gallego-Sala et al., 2018). Bioclimatic envelope models enable estimates of regional peatland area changes of blanket bogs in the United Kingdom (Gallego-Sala et al., 2016; Ferretto et al., 2019) and China (Cong et al., 2020). Process based models provide another way to project potential futures of complex systems under changing boundary conditions. Peatland projections however have mostly focused on peatland area (Alexandrov et al., 2016) and peatland carbon dynamics

(Spahni et al., 2013; Warren et al., 2017; Wang et al., 2018; Chaudhary et al., 2017; Voigt et al., 2019; Swinnen et al., 2019; Chaudhary et al., 2020) independently. A still limited but increasing number of Dynamic Global Vegetation Models (DGVMs) with dynamically determined peatland area (Kleinen et al., 2012; Stocker et al., 2014b; Largeron et al., 2018; Qiu et al., 2018) enables, for the first time, the projection of peatland area and carbon dynamics on a large spatial scale (Qiu et al., 2020). The focus however is still often put on northern boreal peatlands alone (Chaudhary et al., 2020; Qiu et al., 2020).



The dynamic simulation of peatlands is complicated by the non trivial model spinup. Peatland initiation, expansion, and peat carbon accumulation and loss occurred at different times in different regions over the glacial termination and the Holocene as climate and environmental conditions changed. However, peat models are typically spun up uniformly for all regions, over a constant time period, and by applying constant preindustrial climate and environmental (e.g., $CO_2$, land and land use area)

conditions. This common spinup approach does not fully account for the transient and gradual evolution of peatlands, driven and constrained by transient climate evolution, ice sheet retreat and sea level rise (Loisel et al., 2017). In a system with long timescales such as peatlands, the system's history might be a strong determinant of future changes.

Here we present the first combined projection of global peatland area and carbon dynamics. A previously published transient simulation from the Last Glacial Maximum (LGM, 21,000 years before present) to the present (Müller and Joos, 2020) is used

to base the projections on a fully transient spinup. This allows not only to consider all legacy effects of the transient peatland development, but also the consideration of former peatlands in the carbon balance calculation. Committed and future peatland responses to three different future emission and land-use scenarios are investigated using the DGVM LPX-Bern. Uncertainties and drivers are analyzed using multiple climate model forcings and factorial simulations.

## 2 Methods

### 2.1 Model description

All simulations were performed with the Land surface Processes and eXchanges (LPX-Bern) dynamic global vegetation model (DGVM) version 1.4 (Lienert and Joos, 2018). The model setup is mostly identical to Müller and Joos (2020) and the model description is partly adopted from there. LPX-Bern includes an interactive carbon, water and nitrogen cycle and simulates dynamic vegetation composition with plant functional types (PFTs), which compete for water, light and nutrients (Sitch et al.,

2003; Xu-Ri et al., 2012; Spahni et al., 2013). The implementation of permafrost and peatlands as long term carbon stores are based on the LPJ-WHyMe model (Wania et al., 2009a, b) with the addition of dynamic peatland area (Stocker et al., 2014b).

Peatland vegetation is represented by five peat plant functional types (PFTs): *Sphagnum* and flood tolerant graminoids as indicative mostly for high latitude peatlands, and flood tolerant tropical evergreen, decidious tree PFTs, and a flood tolerant C4 type grass, as indicative mostly for tropical peatlands (Stocker et al., 2014b). Carbon cycling in peat soils is based on the

distinction between a lower, fully water saturated slow overturning pool (catotelm; from 0.3 m to 2 m of the soil column) and an upper fast overturning pool (acrotelm; upper 0.3 m of the soil column) with fluctuating water table position (WTP) (Spahni et al., 2013). Decay rates are modulated by temperature in the catotelm and by temperature and WTP in the acrotelm (Wania et al., 2009a). The size and sign of the carbon flux between acrotelm and catotelm is determined by the acrotelm carbon balance. Methane emissions from peatlands are simulated but not part of the analysis in this study.

The area fraction covered by peat in a given grid cell is determined dynamically with the DYPTOP module (Dynamical Peatland Model Based on TOPMODEL) (Stocker et al., 2014b). The TOPMODEL approach (Beven and Kirkby, 1979) is used to predict the monthly inundated area fraction given sub-gridscale topographic information and mean grid cell WTP, averaged over all land classes. The area potentially available for peatlands is then determined by inundation persistency. Peatlands





expand or shrink towards a changing potential extent with a rate of $1\%$ of their current gridcell fraction per year. The gridcell fraction lost during peatland retreat is treated as a separate land class for former peatlands. It inherits the carbon stocks of the shrinking peatland and is subsequently treated in the same way as the mineral soils regarding vegetation, hydrology and carbon cycling. Growing active peatlands first expand on former peatlands inheriting the remaining carbon there. This treatment

prevents carbon dissolution into mineral soils due to area fluctuations.

Different to Müller and Joos (2020) we also consider changing land-use area in our simulations. Land-use area in the model is represented by three specific land classes: pasture, cropland, and urban (Lienert and Joos, 2018). Pastures and croplands have specific vegetation represented by two PFTs each. Changes in land-use area are treated as net changes, where all growing land classes proportionally inherit the carbon, water, and nutrients of all shrinking land classes. A more complex implementation,

which considers gross changes, exists but is not compatible with the peatland module used here (Stocker et al., 2014a). In the absence of gross change information three assumptions were made: 1) Changes within the three land-use classes that do not affect the total land-use area are assumed as shifts between land-use types (e.g. shift from pasture to cropland). 2) Increases in total land-use area reduce all other land classes proportionally, including peatlands. 3) Peatland area that is converted to land-use area can not be reclaimed by expanding peatlands at a later stage. These assumptions are simplifications that fail especially

in areas where peatlands are preferentially targeted for land-use conversion, like e.g. in Indonesia (Dommain et al., 2018; Hoyt et al., 2020) or are subject to restoration efforts after conversion (e.g. Haapalehto et al., 2011; Young et al., 2017). However, given the technical restrictions and the lack of detailed world-wide information about gross land-use changes on peatlands, we think this simplified approach is the most robust.

The above described representation of peatlands in the LPX is a simplification in many respects. The absence of local

processes and information like lateral water flow, local soil features, or influence of animals by grazing and river damming can limit the ability of the TOPMODEL approach to predict peatlands on a regional to local scale. Further, direct human-caused influences such as land use, drainage, or peat mining are only considered in a strongly simplified way. The lack of a distinction and transition between different peatland types like fens, bogs, blanket bogs, or marshes neglects possible differences in the constraints on their formation and evolution. The treatment of acrotelm and catotelm as single carbon pools, and the absence of

strong disturbances such as peat fires, constitute limits on the comparability of the model results to peat core carbon profiles. This simplified representation, nevertheless, has been shown to reproduce peatland area and carbon accumulation well within the observational constraints (Wania et al., 2009a; Spahni et al., 2013; Stocker et al., 2014b, 2017; Müller and Joos, 2020) while using a minimal set of free parameters.

## 2.2 Calculation of peat carbon

Peat carbon can be present not only in soils of active, but also in the soils of former peatlands. Peat may be preserved during peatland conversion and form distinct organic soil layers on non-peatland areas (Lähteenoja et al., 2012; Broothaerts et al., 2014; Xu et al., 2016; Campos et al., 2016; Treat et al., 2019). In the model, subsequent land classes inherit the soil carbon from former peatlands, including peatlands converted to land-use areas. Yet, this peat carbon is mixed within the model's soil pools and cannot be directly distinguished from carbon transferred to soils from more recently established vegetation. It is



however possible to track peat carbon that at one point was sequestered in the catotelm of active peatlands through the soil pools of other land classes using post processing. For this, transient model output for peatland area changes, the decay rates of slow overturning pools, and the carbon input into the catotelm of active peatlands is needed. Area changes are used to transfer carbon between active peatlands, former peatlands, land-use areas and natural vegetation classes. Transient decay rates are used

to decay the carbon in the respective pools. Carbon is thus tracked from its entry into the catotelm of an active peatland until its decay there or in a former peatland or land-use area. This approach can not take account of the acrotelm carbon. However, acrotelm carbon constitutes only a small part of simulated total peatland carbon (5 % at 1975), and we can assume that this carbon at the peat surface is quickly respired after peatland transformation. For the analysis we refer to two different variables related to peat carbon: 1) *peatland carbon* which refers to the carbon stored in the acrotelm and catotelm pools of active

peatlands, and 2) *total peat carbon* which is calculated in post processing and represents all carbon that was at some point accumulated into the catotelm by active peatlands, including carbon in former peatlands or peatlands transformed to land-use areas. After ecosystem transformation, depending on the transition and the conditions thereafter, former peatlands can both see a fast collapse or erosion of carbon stocks (Hoyt et al., 2020; Li et al., 2018) as well as buried peat carbon layers preserved for millennia (Treat et al., 2019). The two carbon variables can be interpreted as two bounding cases to the fate of peat carbon in

former peatlands. Changes in the variable *peatland carbon* can represent a fast emission bounding case where peatland carbon is lost immediately after ecosystem or land-use transformation. The slow emission bounding case, with peat carbon decaying in former peatlands over a long time scale, can be represented by changes in the variable *total peat carbon*. The true fate of peat carbon in former peatlands in most cases will lie somewhere in between these worst and best case scenarios.

## 2.3   Simulation setup

The simulations presented here are a direct continuation of a transient simulation from the Last Glacial Maximum (LGM) to the present, which was discussed in detail in Müller and Joos (2020). This enables future projections starting from a truly transient spinup, including all potential legacy effects of the past 22,000 years. The LGM simulation was run with a model resolution of 2.5° latitude × 3.75° longitude and was forced with $CO_2$ (Joos and Spahni, 2008), temperature, and precipitation fields. Temperature and precipitation anomalies were taken from the transient CCSM3 run TraCE21k (Liu et al., 2009). The TraCE21k

anomalies were imposed on the CRU TS 3.1 (Mitchell and Jones, 2005) base climate from 1960 to 1990. Interannual variability thus came from TraCE21k. Temperature anomalies were calculated as absolute and precipitation anomalies as relative values.

The resolution of the LGM simulation was adopted for the future simulations. This ensures a truly seamless transition between the simulations, without unpredictable effects of down scaling on peatland dynamics.

In the original LGM simulation land-use was not considered as the focus of the study was on the natural development and
evolution of peatlands since the LGM. To integrate a transient history of land-use, the simulation was restarted at the year 1500 with subsequent transient land-use forcing (Hurtt et al., 2020), and otherwise unchanged boundary conditions.

At the year 1975, the midpoint of the base climate period, forcing transitions from TraCE21k to CMIP6 climate anomalies (temperature, precipitation and cloud cover) (O'Neill et al., 2016, see Fig. 1), whereas the base climate remains unchanged. From this point on, simulations are done for each member of a ten member climate model sample (see sect. 2.5). Short





historical simulations from 1975 to 2014 bridge the gap between the LGM simulation and the start of the CMIP6 scenarios with anomalies taken from the CMIP6 historical simulation of the sample climate models.

Starting from the year 2015, simulations corresponding to three different CMIP6 scenarios are run over almost 5.000 years until the year 7000. One strong mitigation (SSP1-2.6), one middle of the road (SSP2-4.5) and one high emission scenario

(SSP5-8.5) were selected to represent the scenario range. The standard CMIP6 scenarios end at the year 2100. To extend the climate further into the future, a detrended version of the last 30 years of each time series was repeated. The trend correction was done per gridcell and month and with respect to the end of the time series. Scenario $CO_2$ forcing was adopted from Meinshausen et al. (2020). Land-use forcing is taken from the Land-Use Harmonization (LUH2) project (see Fig. 2 (g-i), Hurtt et al., 2020).

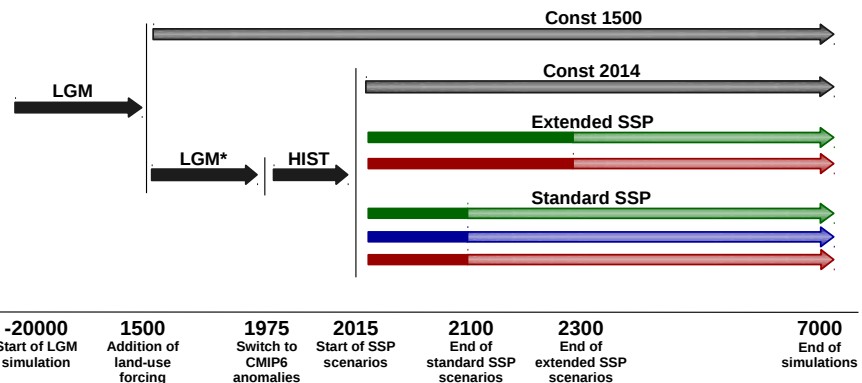

**Figure 1.** Diagramm of the simulation setup. A transient simulation from the LGM to 1975 and with additional land-use forcing after 1500 (LGM*) is followed by short historical simulations until 2014 (HIST) and subsequent standard and extended SSP scenario simulations forced by CMIP6 climate anomalies. After the end of the transient forcing (full arrows), SSP scenarios are continued with constant forcing (transparent arrows). Additional commitment simulations with constant boundary conditions start at 1500 and 2014

CMIP6 also includes extended versions of the scenarios SSP1-2.6 and SSP5-8.5 that range until 2300. To date, however only a handful climate models have run these extended scenarios. In the climate model sample, three out of ten models provide output for these extended scenarios (see sect. 2.5 and Fig. S1). For these climate models we performed additional simulations with transient climate and $CO_2$ forcing until 2300.

To disentangle future changes in peatlands that are induced by changes in climate, $CO_2$ and land-use up to 2014 from those

15 induced by future changes in these drivers, we performed an additional simulation with constant boundary conditions at 2014 levels for each sample member. Here climate forcing was extended with a detrended version of the last 30 years of the historical forcing. Similarly a control simulation was performed with constant boundary conditions after 1500 to show the undisturbed model state. These simulations reveal the committed changes in peatland area and carbon induced by the deglacial changes prior to the preindustrial state at 1500 and the changes over the historical period until 2014, respectively.



## 2.4 Driver contributions

To determine the different driver contributions to the changes in peatland variables, additional factorial simulations were performed for all scenarios and climate model sample members. For each standard simulation, there are five factorial simulations with one of the five transient forcings (temperature, precipitation, cloud cover, $CO_2$, and land-use) kept constant at 2014 respectively.

The driver contribution to the anomaly of peatland variables was determined as the difference between the standard run anomaly and the anomaly in the respective factorial run. The contribution from already committed changes due to past climate and land-use change, was determined as the anomalies in the simulations with overall constant forcing after 2014 (see Sect. 2.3). The residual of the difference between the sum of all contributions and the standard run anomaly was identified as the contributions from non-linear interactions and other factors not considered in the analysis. Cloud cover was found to have only a minimal effect on the considered peat variables in the LPX-Bern and thus for further analysis its contribution was added to the other/non-linear category.

As a second step, driver contributions were classified as driving contributions (same sign as peatland variable anomaly) and dampening contributions (opposite sign as peatland variable anomaly) and re-normalized respectively. Figures show only the driving contributions of the respective positive and negative peatland variable anomalies.

## 2.5 Climate model selection

We chose a sub-sample of ten climate models out of a CMIP6 ensemble of 22 models (see Fig. 2 (a-f) and Fig. S1) that at the time (June 2020) provided monthly output for all necessary forcing variables: precipitation, near surface temperature, and cloud cover and for all considered experiments: historical, SSP1-2.6, SSP2-4.5, and SSP 5-8.5. Climate model output was searched and downloaded from the earth system grid database. One additional model, the CIESM model, had to be excluded from the ensemble as it showed a discontinuity in the precipitation data between the historical and the scenario simulations.

Three models, IPSL-CM6A-LR, MRI-ESM2-0, and CanESM5, were included in the sample a priori as they were the only ones that also provided output for both of the extended SSP1-2.6 and SSP 5-8.5 scenarios (see Sect 2.3). The other seven were chosen for the sample to optimally represent the ensemble as a whole. The optimization targets of ensemble total range, inter quartile range and median, were defined with respect to the anomalies (from 1961-1990 to 2071-2100) in precipitation and temperature as the most important forcings to the LPX-Bern. The optimization was inspired by McSweeney and Jones (2016). 2000 randomly drawn samples were ranked according to the distance of the sample to the targets with normalized scores calculated and averaged over all individual gridcells, months, scenarios, and variables. The rating of the best performing samples was further improved by a careful hand picked combination, resulting in the final sample including the climate models referenced in Table 1.

This sample performs best calculated both over total land area, and over simulated peatland area alone. Over land, temperature anomaly total range, inter quartile range, and median differ between the sample and the full ensemble by 0.25 °C, -0.18 °C, and 0.02 °C respectively, with larger distance at higher emission scenarios. Averaged over simulated peatland area at 1975,



**Table 1.** CMIP6 earth system models used to force the LPX-Bern. Output data was used for monthly precipitation, surface temperature and cloud cover from the 'r1i1p1f1' variant of the respective historical simulations and future scenarios SSP1-2.6, SSP2-4.5, and SSP5-8.5

| Model | Model reference | Data DOI |
|---|---|---|
| CAMS-CSM1-0 | Rong et al. (2018) | Rong (2019a, b) |
| GFDL-ESM4 | Dunne et al. (2020) | Krasting et al. (2018); John et al. (2018) |
| CanESM5 | Swart et al. (2019a) | Swart et al. (2019b, c) |
| EC-Earth3 | Döscher et al. (2021) | EC-Earth Consortium (EC-Earth) (2019a, b) |
| INM-CM5-0 | Volodin et al. (2017) | Volodin et al. (2019a, b) |
| IPSL-CM6A-LR | Boucher et al. (2020) | Boucher et al. (2018, 2019) |
| MPI-ESM1-2-LR | Mauritsen et al. (2019) | Wieners et al. (2019b, a) |
| MRI-ESM2-0 | Yukimoto et al. (2019a) | Yukimoto et al. (2019b, c) |
| KACE-1-0-G | Lee et al. (2020) | Byun et al. (2019b, a) |
| NorESM2-LM | Seland et al. (2020) | Seland et al. (2019a, b) |

the distances are 0.28 °C, -0.31 °C, and 0.03 °C respectively. For precipitation, anomaly total range, inter quartile range, and median differ between ensemble and sample by 9.7 mm, 0.64 mm, and -0.15 mm over all land area and 7.3 mm, 0.54 mm, and -0.49 mm over peatland area respectively. The optimization procedure thus yielded a sub-sample representative of the larger ensemble, although, given the number of possible combinations, optimization could be improved further with further sampling.

## 2.6 Present day model state

There are still considerable uncertainties connected to estimates of the global area covered by peatlands and the amount of organic carbon stored within them. Estimates for northern peatland area, using various methods ranging from inventory based to machine learning, lie between 2.4 and 4.0 million square kilometers (Mkm$^2$) (Yu et al., 2010; Loisel et al., 2017; Xu et al., 2018b; Hugelius et al., 2020). For tropical peatlands, which are still much less studied than northern peatlands, peatland area
estimates have increased in recent years following the discovery of large new peatland complexes, such as in the Congo basin (Dargie et al., 2017), and due to new methodologies trying to account for potentially undiscovered peatlands (Gumbricht et al., 2017). Earlier estimates of tropical peatland area thus range from 0.37 to 0.44 Mkm$^2$ (Yu et al., 2010; Page et al., 2011) and more recent estimates from 1.0 to 1.7 Mkm$^2$ (Gumbricht et al., 2017; Xu et al., 2018b). Peatland areas simulated by LPX at the year 1975, the end of the transient simulation from the LGM, are within the range of literature estimates with a global, northern
(>30°N), and tropical (30°S to 30°N) peatland area of 3.8 Mkm$^2$, 2.8 and 1.0 Mkm$^2$ respectively. Global peatland area is shifted more towards the tropics as in most estimates. However, most mayor peatland complexes seen in global peatland maps e.g. PEATMAP (Xu et al., 2018b, see Fig. 3), are captured well. Mayor regional differences exist in Africa, where LPX-Bern fails to simulate the large Congo Basin peatland complex. Peatland area is also underestimated in northern Europe. In North America the model overestimates peatland area in Alaska and Quebec and underestimates peatland extent in western Canada.



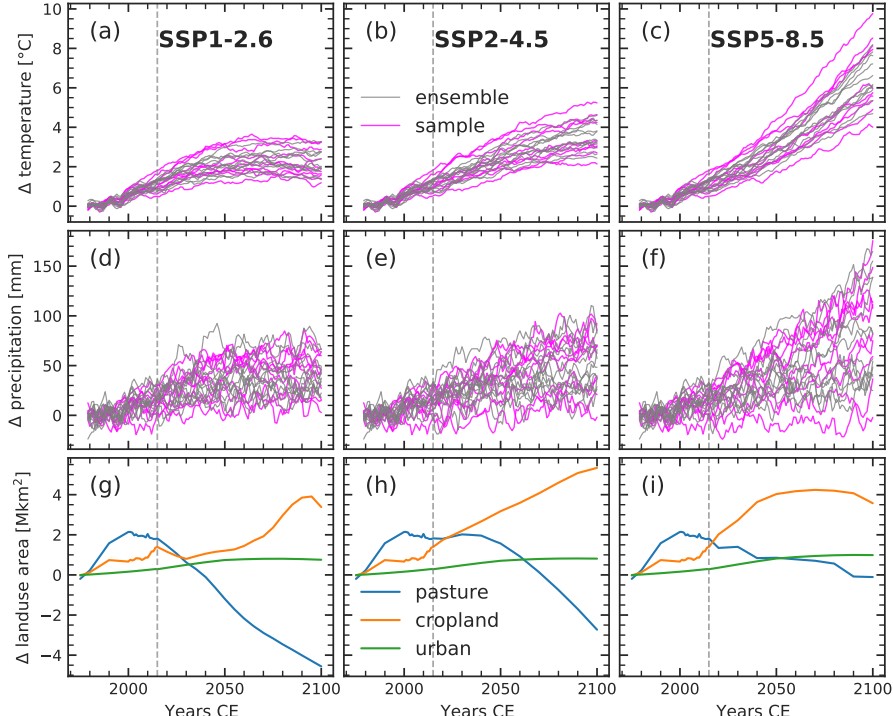

**Figure 2.** Global surface air temperature (a-c), precipitation (d-f) and land-use area (g-i) anomalies to the 1961-1990 average for three CMIP6 scenarios: SSP1-2.6 (a,d,g), SSP2-4.5 (b,e,h), SSP5-8.5 (c,f,i). Magenta lines show anomalies of the climate model sample applied to force LPX-Bern versus the rest of the CMIP6 ensemble in grey. Dashed vertical gray lines show the year 2015 from which the future scenarios diverge. Pasture, cropland, and urban land-use areas amount to 31.2, 14.4 and 0.3 Mkm$^2$ during the 1961-1990 baseline period, respectively

Estimates of global peatland carbon are directly dependent on peatland area estimates and thus also come with a large uncertainty range. Northern peatlands have been estimated to store 270 to 604 gigatons of carbon (GtC), using various methods and area estimates (see Yu (2012) and Yu et al. (2014) for a review). For tropical peatlands estimates of organic carbon storage range from 44 to 92 GtC in earlier estimates (Yu et al., 2010; Page et al., 2011) and increase as a result of larger assumed areas

5    in recent estimates from 70 to 288 GtC (Dargie et al., 2017; Ribeiro et al., 2021). At the year 1975, peatlands simulated by the LPX-Bern have accumulated about 441 GtC of soil organic carbon globally. From this, northern and tropical peatlands make up 319 GtC and 121 GtC respectively. Simulated carbon stocks thus lie within the literature estimates, however, again with a distribution shifted more towards the tropics than most estimates suggest.

Throughout the 22,000 years of the transient simulation up to the year 1975, peatland area was highly dynamic with today's

10   peatlands gradually expanding, but also large paleo peatlands vanishing over time (Müller and Joos, 2020). There, similarly as described in section 2.2, the carbon from former peatlands was tracked through subsequent land classes until its decay. At 1975 LPX-Bern gives a total of 195 GtC of peat carbon left over from former peatlands on land, with 165 GtC in northern



latitudes and 28 GtC in the tropics. Total peat carbon is thus simulated to be 612 GtC. Very little is known about the amount and location of peat left over and buried from former peatlands, although various deposits have been found (Treat et al., 2019). Peat carbon lost due to past or future sea level rise is not considered in this study.

Unlike the original LGM simulation in Müller and Joos (2020), here land-use and land-use change is considered since the
5    year 1500, with land-use areas being able to expand onto peatlands (see Sect. 2.3). Up to 1975 about 0.42 Mkm$^2$ of peatland area and 50 GtC of peatland carbon are lost from active peatlands due to land-use change in the simulation. Carbon loss is reduced to about 5 GtC when considering total peat carbon, which does not assume an immediate loss but a slow decay in former peatlands. Leifeld et al. (2019) estimate that about 0.51 Mkm$^2$ of peatland area and $22 \pm 5$ GtC of peatland carbon was lost globally from 1850 to 2015 due to drainage and landuse conversion of peatlands, which is in rough agreement to the
10   simulated values (see also sect. 3.1.1).

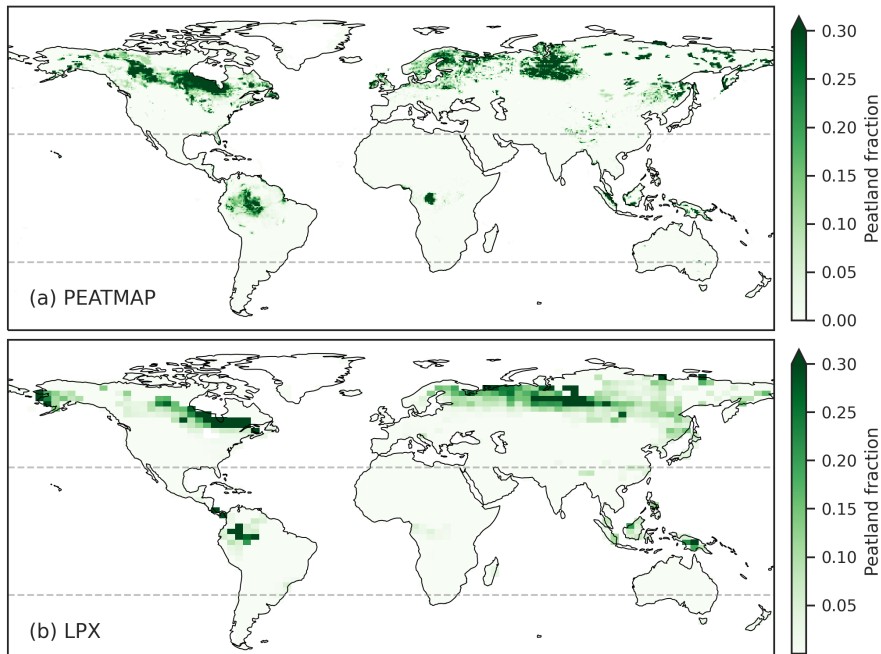

**Figure 3.** Global peatland area fraction as (a) estimated by PEATMAP (Xu et al., 2018b), shown here in a $0.5° \times 0.5°$ gridded version and (b) simulated by LPX at the year 1975 after a transient simulation from the Last Glacial Maximum





**Table 2.** Median and corresponding inter quartile range (IQR) of relative anomalies for simulated peatland area, peatland carbon and total peat carbon as defined in section 2.2. Medians and the IQR are given relative to 1995-2014 averages. IQR boundaries are listed in Table S1. Values are rounded to integer values. Northern latitudes are defined as >30°N and tropical latitudes as 30°S-30°N. For the reference period, simulated median peatland area, carbon and total peat carbon is 3.7 Mkm$^2$, 423 GtC and 611 GtC for global, 2.7 Mkm$^2$, 301 GtC and 500 GtC for northern, and 1 Mkm$^2$, 121 GtC and 142 GtC for tropical peatlands respectively

| | Δ peatland area [%] | | | Δ peatland carbon [%] | | | Δ total peat carbon [%] | | |
|---|---|---|---|---|---|---|---|---|---|
| | 2100 | 2300 | 3500 | 2100 | 2300 | 3500 | 2100 | 2300 | 3500 |
| **Global** | | | | | | | | | |
| Committed | -3 (4) | -4 (4) | +8 (5) | -7 (3) | -9 (3) | +4 (6) | -0 (0) | -1 (1) | +1 (4) |
| SSP1-2.6 | -7 (5) | -4 (17) | +16 (22) | -14 (8) | -12 (21) | +7 (26) | -0 (1) | -0 (3) | +3 (14) |
| SSP2-4.5 | -11 (10) | -23 (19) | 0 (27) | -19 (11) | -33 (22) | -17 (31) | -0 (1) | -2 (4) | -4 (18) |
| SSP5-8.5 | -14 (9) | -29 (13) | -2 (27) | -22 (10) | -43 (16) | -29 (20) | -1 (1) | -5 (6) | -17 (21) |
| **Northern** | | | | | | | | | |
| Committed | -8 (4) | -11 (6) | -4 (5) | -10 (4) | -12 (4) | -1 (4) | +0 (0) | -0 (0) | -0 (2) |
| SSP1-2.6 | -15 (12) | -18 (28) | -8 (25) | -19 (10) | -18 (30) | -5 (35) | -0 (1) | -1 (3) | -2 (14) |
| SSP2-4.5 | -22 (15) | -41 (29) | -31 (31) | -26 (15) | -47 (30) | -37 (37) | -0 (1) | -3 (4) | -14 (19) |
| SSP5-8.5 | -28 (9) | -61 (20) | -54 (30) | -32 (12) | -65 (16) | -61 (27) | -1 (1) | -7 (6) | -32 (25) |
| **Tropical** | | | | | | | | | |
| Committed | +9 (7) | +14 (10) | +37 (8) | -0 (3) | +2 (5) | +21 (8) | -0 (1) | -1 (1) | +5 (4) |
| SSP1-2.6 | +14 (11) | +27 (19) | +66 (26) | +2 (3) | +5 (8) | +32 (21) | +0 (1) | -0 (5) | +13 (16) |
| SSP2-4.5 | +14 (10) | +34 (15) | +84 (19) | -1 (2) | +2 (6) | +32 (13) | -1 (1) | -1 (4) | +13 (10) |
| SSP5-8.5 | +19 (9) | +60 (34) | +143 (31) | -0 (4) | +7 (13) | +56 (31) | -1 (1) | +1 (9) | +28 (28) |

# 3 Results and Discussion

## 3.1 Historic and committed changes

### 3.1.1 Historic 1975-2014

The gap between the end of the transient LGM run in 1975 and the beginning of the future scenarios in 2015 is bridged by short

5 historical simulations. These are forced with climate anomalies from the ten sample climate models (see sect. 2.3). During this short period climate anomalies already drift apart substantially between the different climate models (see Fig. 2). Differences in climate forcing are propagated to differences in peatland responses. Averaged over 1995-2014 simulated global peatland area varies between 3.6 Mkm$^2$ - 3.8 Mkm$^2$. Legacy effects and accelerating climate change lead to a reduction in peatland area with respect to 1975 in most simulations, resulting in a median of 3.7 Mkm$^2$. A part of this reduction (median: -0.1 Mkm$^2$) is

10 also attributable to an increase in land-use area, which claims an additional 0.06 Mkm$^2$ from 1975 to 2014.





The respective carbon stored in global peatlands is simulated to be 423 (419-432) GtC, with changes mostly a result of the peatland area changes. Changes in total peat carbon, including carbon in former peatlands (see Sect. 2.2), also are mostly negative but small, with global peat carbon stocks at 611 (612-610) GtC.

### 3.1.2 Committed 2015-2300

Past changes in climate and land-use have long lasting effects on global peatlands that are superimposed on changes induced by future disturbances. To disentangle the effects of past and future changes in drivers, simulations with constant 2014 boundary conditions were made for each climate model sample member. These "committment" simulations reveal the delayed response to disturbances in the past and thus represent the committed changes independent of the future scenario for climate, land use and $CO_2$ (Fig. 4, Fig. 5, and Fig. S2).

In most committment simulations, global peatland area continues to decrease and reaches a new equilibrium until 2300. Gross changes reveal, however, also regions of local peatland expansion (Fig. 4). North eastern Canada, northern Europe, and east Asia are regions with large losses, whereas north west Canada, north east Asia, and south east Asia see an increase in peatland area up to 2300.

   Peatland carbon decreases together with global area in most simulations, with new peatland area showing lower carbon

density as lost areas. Total peat carbon, depending on the overall balance of accumulation and decay rather than on peatland area dynamics, is changing only slightly but is declining in eight out of ten simulations. Taken together, the simulations suggest a small to moderate peat carbon loss to the atmosphere over the next 300 years given 2014 conditions. Uncertainties however are large. The spread between the simulations increases significantly after 2014 despite boundary conditions being kept constant. At the year 2300, the simulated global peatland area anomaly relative to 1995-2014 averages ranges from -13 to

+4 %, with a median of -4 % and inter quartile range (IQR) from -6 to -2 % (Table 2). Global carbon stored in active peatlands and global total peat carbon are simulated to change by -9 (total range: -16 to -0; IQR: -10 to -7) % and -1 (-2 to +1; -1 to -0) % respectively. The increasing uncertainty highlights how relatively small differences in forcing can propagate and result in large long term ecosystem and carbon cycle uncertainties.

### 3.1.3 Committed after 2300

Some simulated peatland responses to historic changes in climate and land-use are delayed even beyond 2300. Between about 2700 and 3500 all simulations see a rapid peatland expansion. At 3500 gobal peatland area anomaly compared to 1995-2014 averages is +8 (-1 to +16; +5 to +11) %. With that peatland area is simulated even larger than at present, however, with a dramatically shifted global and regional distribution. The delayed peatland expansion is limited to the northern highest latitudes and the tropics. The resulting peatland distribution at 3500 shows loss of sizable parts of today's northern peatlands with new

peatlands partly expanding into permafrost regions and the tropics.

   Carbon accumulation within old as well as newly formed active peatlands continues over millennia reaching a global peatland carbon stock of +4 (-4 to +16; +1 to +8) % at the year 3500 compared to 1995-2014 averages, illustrating a large long term accumulation potential (Fig. S2).





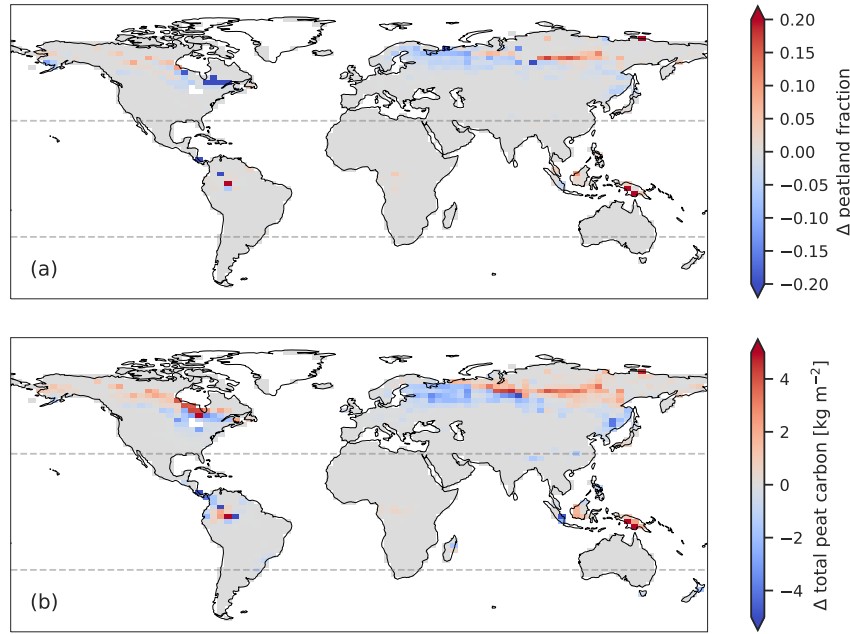

**Figure 4.** Median anomalies for the simulations with constant 2014 forcing at 2300 in (a) peatland area fraction and (b) total peat carbon as defined in section 2.2. Anomalies are calculated between 20 year averages spanning 1995-2014 and 2291-2310

For global total peat carbon, the expansion in peatland area also results in a trend reversal in most simulations. The large accumulation in the newly established peatlands helps to shift the balance from decay dominated to accumulation dominated. At 3500 total peat carbon is simulated at +1 (-5 to +7; -1 to +2) % compared to 1995-2014 averages and continues to increase with continued accumulation until the end of the simulation.

### 3.1.4 1500 control simulation

An additional simulation with constant 1500 CE boundary conditions, and thus with only limited land-use and no industrial climate change, shows, that without mayor disturbance, peatland area remains stable, with only a small increase of 2 % over the whole 5500 years of simulation (Fig. 5 and S2). Carbon shows a stronger positive trend reflecting the still large accumulation potential of undisturbed global peatlands, with an increase of 22 % and 17 % for peatland and total peat carbon respectively.

Millennium scale accumulation rates however are larger in simulations with constant 2014 boundary conditions, due to higher productivity in high latitudes and newly emerging peatlands. This indicates, that despite an initial loss, peat carbon storage under 2014 conditions could exceed storage under 1500 conditions, but only after millennia of ecosystem transformations and renewed carbon accumulation. Another study investigating the fate of permafrost peatland carbon in the circum-Arctic region comes to similar conclusions about future long term storage capacity of peatlands (Swindles et al., 2015).



### 3.2 Future Projections

#### 3.2.1 Standard scenarios

#### 3.2.2 2015-2300

The standard CMIP6 scenarios provide transient climate anomalies from 2015 to 2100 after which boundary conditions are
5 held constant. Until the end of the century global peatland area, peatland carbon, and total peat carbon are simulated to decline
(Table 2). This decline is larger than committed changes alone and increases with increasing scenario-based emissions for all
three variables (Fig. 5 and Fig. 6). These results suggests a clear relationship between future emissions pathways and resulting
peatland area and carbon losses until the end of the century.

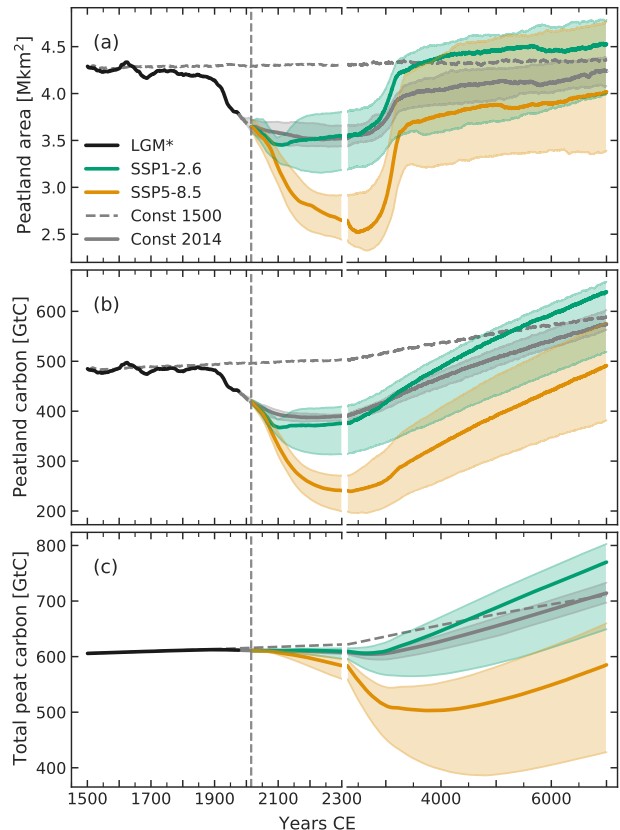

**Figure 5.** Simulated continued transient evolution after the LGM run (LGM*) of (a) global peatland area, (b) global peatland carbon, and
(c) global total peat carbon as defined in section 2.2 under the SSP1-2.6 and SSP5-8.5 scenarios and under constant 1500 and 2014 forcing.
SSP2-4.5 is not plotted to increase readability. Lines and shading for SSP1-2.6, SSP5-8.5 and constant 1500 forcing show sample medians
and interquartile ranges respectively. The dashed vertical line indicates the year 2014. Note the change in the time axis after the year 2300

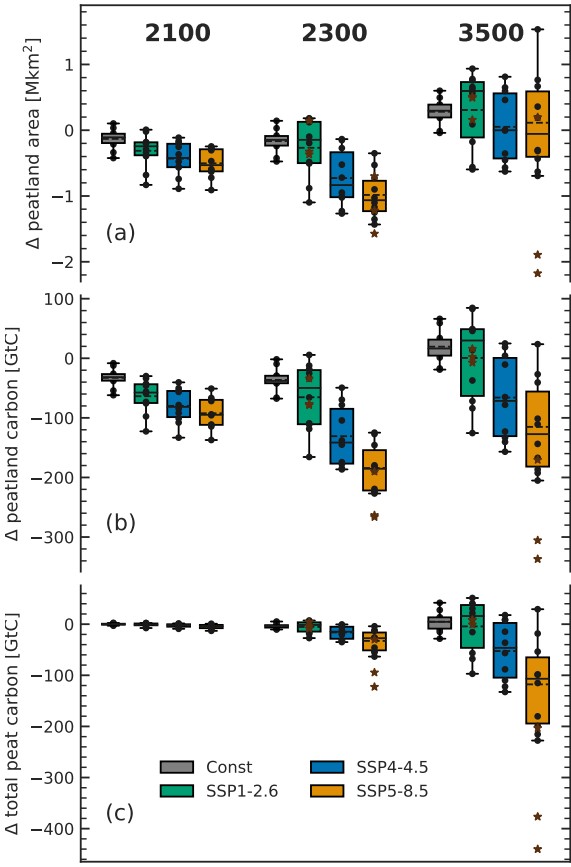

**Figure 6.** Boxplots of anomalies in (a) global peatland area, (b) global peatland carbon, and (c) global total peat carbon as defined in section 2.2 for simulations under three future scenarios and constant 2014 forcing. Black dots indicate the individual simulations forced with different climate model anomalies. Boxes indicate the interquartile range, whiskers the total range, solid lines the median, and dashed lines the mean. Brown stars indicate additional simulations forced with extended versions of scenario SSP1-2.6 and SSP5-8.5. Anomalies are calculated between 20 year averages (2091-2110, 2291-2310, and 3491-3510) and the reference period 1995-2014

From 2100 to 2300 global peatland area, peatland carbon, and total peat carbon continue to decrease for scenarios SSP2-4.5 and SSP5-8.5 despite constant boundary conditions after 2100 (Table 2). Only under the strong mitigation scenario SSP1-2.6, most simulations show an increase in peatland area and partly in carbon compared to 2100. Medians are similar to the simulations under constant 2014 forcing. However the uncertainty, represented by the spread between the simulations, is larger
5 in the SSP scenarios than in the commitment simulations. For all scenarios this uncertainty increases with time.

Spatial anomaly patterns at 2300 for the strong mitigation scenario SSP1-2.6 (Fig. 7), are similar to the committed changes (Fig. 4). Regions of peatland area loss are north eastern Canada, northern Europe, central Russia and east Asia and peatland





area increases can be found in north west Canada, north east Asia, and south east Asia. Losses and gains are further amplified with respect to the committed changes. Thus, the increase in global area after 2100 in SSP1-2.6 is not due to a recovery of lost peatlands, but rather due to a stronger increase of peat area in the regions of local peatland expansion.

The higher the scenario-based emissions the more extensive the regions of peatland area and carbon loss in the northern high latitudes become and the more reduced are regions of gains. In the high emission scenario SSP5-8.5, losses dominate most of the northern high latitudes (Fig. 8). Small regions of area gains remain in north east Asia, however with weakened expansion compared to lower emission scenarios. Northern peatland area is simulated to reduce by 18, 41 and up to 61 % until 2300 under the SSP1-2.6, SSP2-4.5 and SSP5-8.5 respectively. These results suggest that large parts of today's northern peatlands might be at risk under future climate change. In the tropics this trend is reversed, with area expansion in South America and south east Asia amplified under higher emission scenarios. Although net area gains in the tropics are substantial, carbon anomalies remain small and at 2100 only positive for the strong mitigation scenario, indicating a strong concurrent increase in heterotrophic respiration.

Taken together these results suggest a likely net loss of global peatland area as well as carbon until the end of the century, driven mostly by northern peatlands, even under the strongest mitigation scenario and continued net loss up to 2300 for the scenarios SSP2-4.5 and SSP5-8.5. This is in contrast to another recent modeling study investigating future northern peatland area and carbon dynamics. Qiu et al. (2020) used the ORCHIDEE-PEAT DGVM model, with similar TOPMODEL driven peatland area dynamics as the LPX-Bern and forced by IPSL-CM5A-LR and GFDL-ESM2M model climate to simulate northern peatland dynamics from 1861 to 2099. They found a strong positive trend in northern peatland area and together with a sustained sink also in peatland carbon over the whole historical period and the two investigated scenarios RCP2.6 and RCP6.0. They identify the main driver of this trend as an internal feedback between gridcell water table position and peatland area, which is independent of the climate forcing. This feedback is also part of the LPX-Bern implementation, however, here it does not lead to a strong sustained historical or future increase in northern peatland area, illustrated by the relatively stable control simulation under constant 1500 boundary conditions (Fig. 5). On the contrary here it amplifies the simulated negative trend (see sect. 3.3). One of the main reason for the different behaviours might be the spinup procedure which is very different in both cases. Whereas Qiu et al. (2020) used an idealized spinup with constant climate conditions, the spinup in this study corresponds to a full transient simulation. Other factors could be the different parametrizations and implementation details. Despite the differences in projected net peatland area trends, there are also regional agreements. Qiu et al. (2020) found central and northern Europe to be regions of future peatland loss, especially given the warmer IPSL-CM5A-LR forcing, and north eastern Asia a region of particular strong peatland expansion, partly matching regional patterns presented here. For central Russia, the simulated dynamics partly agree with another modeling study. Alexandrov et al. (2016) projected potential future peatland area in Western Siberia using an impeded drainage model and MPI-ESM climate anomalies. They found a strong increase in potential area north of 60°N and a strong decrease south of 60°N. A similar response pattern can be found in the simulations forced with MPI-ESM1-2-LR climate anomalies and weaker also in the sample medians, however only up to the SSP2-4.5 scenario after which losses dominate over Western Siberia.

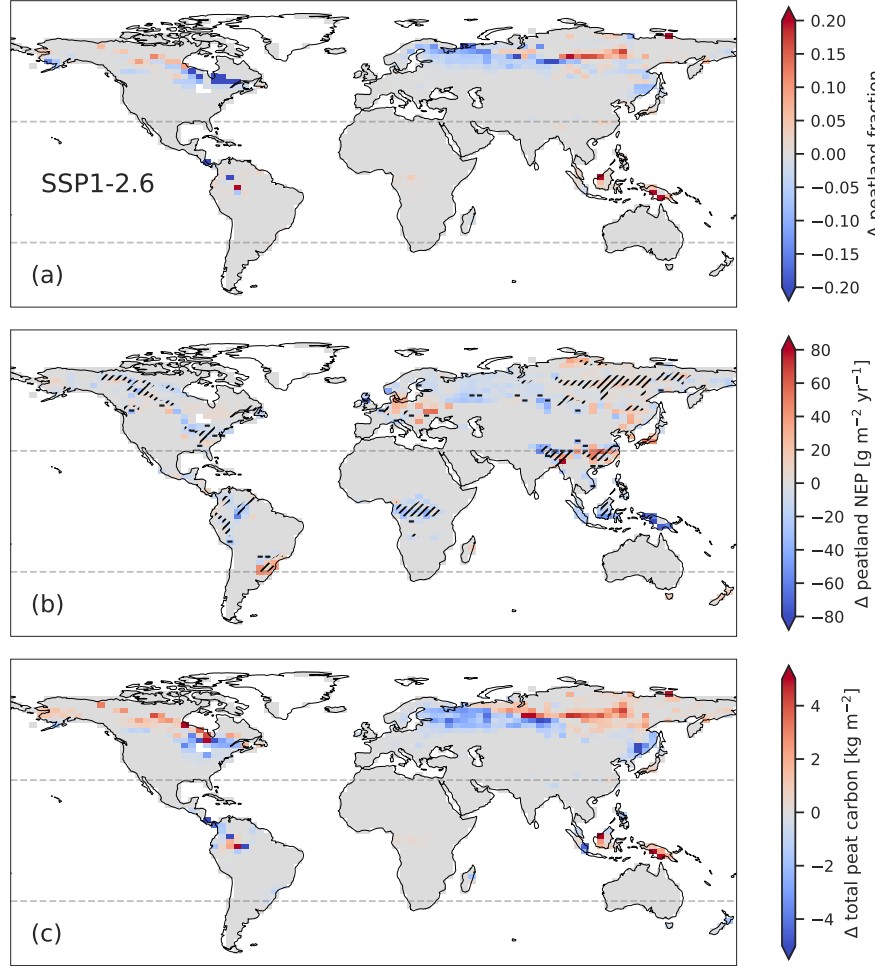

**Figure 7.** Simulated median anomalies under the SSP1-2.6 scenario at the year 2300 in (a) peatland area fraction, (b) peatland NEP, and (c) total peat carbon as defined in section 2.2. Anomalies are calculated between 20 year averages spanning 1995-2014 and 2291-2310. Gridcells where NEP becomes negative are marked with a minus. Hatched areas in (b) indicate a positive area anomaly, which in the model can lead to an increase in NEP through the dilution of soil carbon and a corresponding reduction in soil carbon respiration per area

    Peatland area dynamics translate directly and indirectly into the simulated carbon dynamics. In the strong mitigation scenario, the simulated net loss of northern peatland carbon and total peat carbon is mainly a result of the northern peatland dynamics, rather than of declining carbon accumulation rates. Figure 7 (b) shows that the net ecosystem production (NEP) of active peatlands, which represents the net carbon uptake from the atmosphere per year, changes only slightly until 2300 with

5   decreases throughout the tropics and in parts of the northern latitudes. Regional increases in NEP are simulated in central and eastern Europe, and east Asia. Similar is true for the SSP2-4.5 scenario, however with slightly larger decreases in the north-


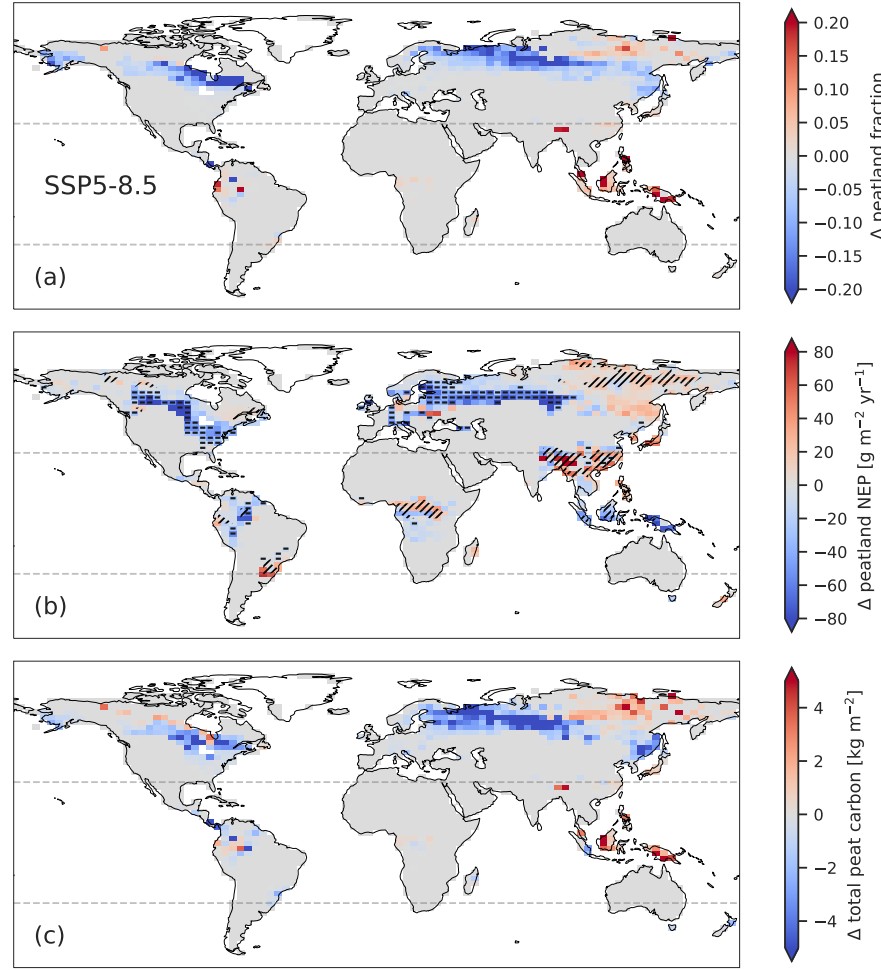

**Figure 8.** As Fig. 7 but for the SSP5-8.5 scenario

ern mid-latitudes. Under the high emission scenario SSP5-8.5, simulated NEP decreases strongly in North America, Europe and western Asia, with most mid- to high-latitude active peatlands turning from a carbon sink to a carbon source and thus contributing directly to the net carbon loss. Regionally NEP increases are simulated again mostly in east Asia, with larger increases compared to SSP1-2.6. (Fig. 8 (b)). It has to be noted that in case of regional peatland expansion NEP might increase

5 independent of environmental drivers, simply due to the dilution of soil carbon. The results are in broad accordance with a previous study conducted with an older version of the LPX-Bern. Spahni et al. (2013) conducted transient northern peatland simulations from the LGM up to 2100 with prescribed peatland area using CMIP5 future climate anomalies together with the LPX-Bern version 1.0. They found mean northern peatland NEP to slightly decrease over time under the RCP 2.6 scenario and strongly decrease under RCP 8.5. Qiu et al. (2020), however, simulate net NEP of northern peatlands to increase slightly and





peak mid century before a decline back to roughly 2005 levels at 2099. Regionally, Qiu et al. (2020) project NEP to decline in western Canada, western Europe and the China–Russia border especially under the RCP6.0 scenario, however only matching western Canada as a region of NEP decline simulated by LPX-Bern. In both models, northern peatland productivity and soil carbon respiration increase concurrently, however balanced slightly differently. Chaudhary et al. (2020) investigated past and

future CARs of northern high latitude peatlands up to 2100 using a dynamic vegetation model and similarly found net increases in simulated carbon accumulation rates (CARs) until the mid century and declining rates thereafter, most pronounced under the RCP8.5 scenario. They found Siberian and highest latitude peatlands to potentially increase their CARs where as northern European and north American mid-latitude peatlands were most vulnerable to carbon sink decreases or even carbon loss, roughly matching regions of NEP increases and decreases simulated by LPX-Bern. Gallego-Sala et al. (2018) used data-derived rela-

tionships between CARs and climate variables for global future projections. They find a latitudinal-dependent response, with CARs increasing continuously until 2300 in high latitudes and decreasing in low latitudes. Under a high emission scenario, CARs switched in mid latitudes from an increasing to a decreasing trend with rising temperatures. Changes in peatland NEP simulated by the LPX have a less latitude determined pattern, but also project carbon uptake to decrease most strongly in the tropics and large parts of the mid latitudes, with widespread switching to net carbon sources. Large increases in the northern

high latitude peatland NEP however are only simulated in east Asia and parts of eastern Europe.

### 3.2.3  After 2300

Similar to the committed changes, the future scenarios see a delayed rapid expansion in global peatland area between about 2700 and 3500 CE (Fig. 5, Fig. 6, and Table 2). Regionally this expansion is dominated by new peatlands in tropical south Asia, with smaller contributions in the highest northern latitudes. The expansion in south Asia increases in magnitude under

increasing emission scenarios. This leads to a large overlap between the uncertainty ranges of the different scenarios after 3500. The median of global peatland area anomaly in 2300, however, is substantially higher for SSP1-2.6 than for the other scenarios and for constant 2014 climate. Medians of global peatland area in the higher emission scenarios SSP2-4.5 and SSP5-8.5, remain below the levels given by 2014 boundary conditions for the whole time of the simulation, despite the large delayed expansion.

Carbon storage in old and new active peatlands increases continuously after 2300, with similar rates for all scenarios, preserving the large differences in total peatland carbon stocks formed in previous centuries. The peatland expansion after about 2700 also results in a trend reversal in total peat carbon which continues to increase thereafter. Rates of global carbon sequestration are larger than rates under constant 1500 or 2014 conditions, most pronounced in the SSP1-2.6 scenario. Towards the end of the simulation, total peat carbon storage in SSP1-2.6 exceeds carbon stocks under constant 1500 conditions. This

reflects higher mean productivity of the remaining and newly formed peatlands under moderately warmer and wetter conditions and with a higher atmospheric $CO_2$ concentration.

Taken together, the long term response of global peatlands to future climate change, suggest that under strongly limited future climate change and after negative effects dominating over centuries, potential gobal peatland area and peat carbon could increase compared to today, and even to pre-industrial levels on a millenial timescale. Higher emission scenarios however





show a negative effect on global peatland area and a reduced peat carbon storage potential persisting for millennia compared to constant 2014 conditions. Uncertainties towards the end of the simulations, indicated by the sample spread, however become very large.

### 3.2.4 Extended scenarios

The assumption of stable climate and atmospheric $CO_2$ levels over millennia after 2100 is a highly idealized one, intended to reveal delayed responses and long lasting effects in the slow reacting peatland system. Depending on the future emission pathway, global temperature and atmospheric $CO_2$ is expected to either decline, stabilize or dramatically increase beyond 2100 (see Sect. 3.2.4). Continued ocean and land uptake of $CO_2$ and heat will shape the future climate, see level and atmosphere for centuries to millennia after greenhouse gas emissions stop (Frölicher and Joos, 2010; Zickfeld et al., 2013; Frölicher et al., 2014; Clark et al., 2016). Additionally, possible tipping points in the earth system could abruptly change the trajectory of the climate system (Lenton et al., 2008). Learning about the long term responses to disturbances of key parts of the climate system, such as peatlands and their large carbon stocks, is an important part of understanding future earth system responses as a whole.

To additionally investigate the effect of transient boundary conditions from 2100 to 2300, additional simulations were performed for three of the ten sample models that provided climate output for the extended scenarios SSP1-2.6 and SSP5-8.5 (see Sect. 2). Results are compared to simulations with standard scenarios in Figures 9 and 6. For the extended SSP1-2.6 scenario, global mean temperatures begin to decline again after peaking at about 2100 together with atmospheric $CO_2$ concentrations (Fig. S1). This leads to a weakening of the long term peatland response compared to constant 2100 conditions in all three simulations. Simulations forced with climate anomalies from the IPSL and CanESM models, with relatively large climate sensitivities, show negative global peatland area and carbon variable anomalies up to 2100. After 2100, the extended scenario mitigates the external pressure and long term losses in peatland area and carbon are lower than under continued 2100 conditions. The simulation forced with MRI climate anomalies, with a relatively small climate sensitivity, on the other hand sees a positive global peatland area anomaly at 2100 and thus shows less global peatland area and long term peatland carbon storage under the extended scenario, compared to constant 2100 conditions.

The extended SSP5-8.5 scenario forces global temperatures and atmospheric $CO_2$ to increase drastically until 2300 (Fig. S2). Mean global temperatures over land reach 24-34 °C at 2300 with an atmospheric $CO_2$ concentrations of 2162 ppm. Under these extreme conditions all simulations show a reduction in global peatland area and carbon compared to constant 2100 conditions. The simulations forced with the IPSL and CanESM models lose practically all high- and mid-latitude peatlands as well as the Amazon basin complex until the end of the simulation. This results in a peatland carbon and total peat carbon reduction of about 50-60 %. Old and newly established peatlands that remain until the end of the simulations are mostly located in tropical south Asia, south east Asia and coastal regions of South America. The responses in the simulation forced with climate anomalies from the less sensitive MRI model are less extreme but also show a significant reduction in global peatland area and carbon compared to constant 2100 conditions.

The sub-sample of climate models that provided the extended scenario output is not representative of the whole sample or the CMIP6 ensemble as a whole. The additional simulations however make clear, that the consideration of transient climate after





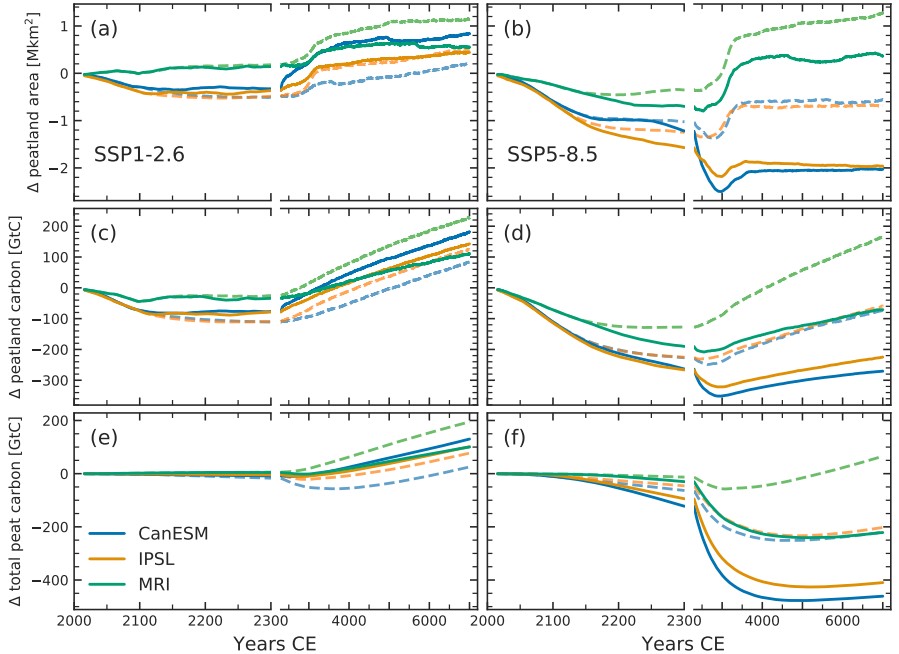

**Figure 9.** Simulated anomalies of (a)-(b) global peatland area, (c)-(d) global peatland carbon, and (e)-(f) global total peat carbon as defined in section 2.2 for three models providing output for the extended future scenarios SSP1-2.6 (a,c,e) and SSP5-8.5 (b,d,f). Solid lines show simulations with extended scenario forcing, evolving transiently until 2300. Dashed lines show simulations with standard scenario forcing, transient only until 2100. Note the change in the time axis after the year 2300

2100 can change simulated long term peatland responses strongly, depending on the emission pathway. As the 21st century becomes ever shorter, the main focus of future projections and climate policy remains on the next few decades up to 2100. To better understand the long term effects of past and future emissions on global peatlands and to assess their potentially large feedbacks on future climate, the horizon of the future must be expanded beyond 2100. Scenarios extended to 2300 should be
5 elevated to standard practice for future climate projections.

### 3.3 Driver contributions

A factorial analysis was used to attribute the positive and negative changes in peatland variables to individual forcing drivers (see sect. 2.4). Figure 10 shows the calculated mean driver contributions to the global gross positive and negative anomalies in peatland area, peatland carbon, and total peat carbon.
10     Increases in peatland area up to 2300, both in the high and low latitudes, are driven mostly by committed changes (constant 2014 conditions) and an increase in regional precipitation. In northern permafrost regions, this is further strengthened given strong mitigation and moderately rising temperatures resulting in longer growing seasons and larger water retention (Fig. A1). Suggested pathways of permafrost peatlands after thaw are still debated, but include rapid degradation (Avis et al., 2011;





Turetsky et al., 2020), collapse followed by long term recovery (Jones et al., 2017; Magnússon et al., 2020; Swindles et al., 2015), to increased carbon accumulation (Estop-Aragonés et al., 2018), depending multiple factors such as thaw velocity and local hydrology.

Simulated peatland area losses are driven mostly by committed changes and increasing temperatures. Higher temperatures

lead to an increase in evapotranspiration, especially in boreal peatlands (Helbig et al., 2020b), and thus a decrease in the regional water balance which is not compensated despite potential concurrent increase in annual precipitation. This corresponds to the already observed decade to century long drying trends in northern Europe (Swindles et al., 2019; Zhang et al., 2020) and eastern Canada peatlands (Pellerin and Lavoie, 2003; Pinceloup et al., 2020; Beauregard et al., 2020), regions of large simulated committed area loss, which are found to result in negative effects on carbon accumulation rates and strong trends of

woody encroachment. These trends are expected to continue and amplify under future climate change.

In the LPX-Bern, a decreasing water balance can lead to a positive feedback on the retreating water table. A long term draw down of the mean gridcell water table leads to a reduction in peatland area, which in turn reduces the mean gridcell water table further. In some cases, this can lead to a much larger reduction in peatland area as would be the result of the initial disturbance. In reality a lot of complex and often still poorly understood both positive and negative hydrological feedbacks

control peatland water table position in response to disturbances (Morris et al., 2011; Waddington et al., 2015). The simplified structure of DGVMs like the LPX-Bern cannot mirror these complex interactions and thus this strong internal feedback needs to be interpreted with caution. However, although peatlands, in some cases, have been found to be relatively resilient with respect to limited disturbances (Cole et al., 2015; Swindles et al., 2016; Page and Baird, 2016), there could be possible tipping points that could lead to fast vegetation and ecosystem transitions under strong persisting disturbance (Eppinga et al., 2009;

Heijmans et al., 2013; Page and Baird, 2016).

Peatland carbon and total peat carbon dynamics up to 2300 are dominated by committed and temperature driven losses. The decline in peatland area directly reduces global peatland carbon and indirectly affects the total peat carbon balance by reducing overall accumulation. At the same time, global decay in active as well as former peatlands is increased by the higher temperatures. Peatland NEP in the northern high latitudes is also driven in large parts by committed changes and increasing

temperatures which can both increase or decrease NEP given the balance between respiration and productivity and their effect on permafrost (Fig. A1 and A2). Increasing $CO_2$, precipitation, and non linear interactions between the drivers have strong positive effects on NEP in east Asia. In the tropics, the negative trend in NEP is mostly driven by the higher temperatures and non linear effects.

Up to 2300, land-use change and atmospheric $CO_2$ have a comparatively small impact on the global scale. Increasing $CO_2$

concentrations have a positive effect on carbon accumulation which is progressively larger with increasing emission scenarios. This also translates into a moderate peatland area gain at 2300. Effects of land-use change are negative for all peatland variables, but remain small. One reason for this small impact, might be that only net land-use area increases are considered to affect peatlands in the model. The global net increase in land-use area, however, is much smaller in the future scenarios, as for the historic period (sect. 2.6). Especially in regions where peatlands are directly targeted for land-use conversion, such as in





Indonesia (Dommain et al., 2018; Hoyt et al., 2020), our approach might significantly underestimates the negative effect of land-use change.

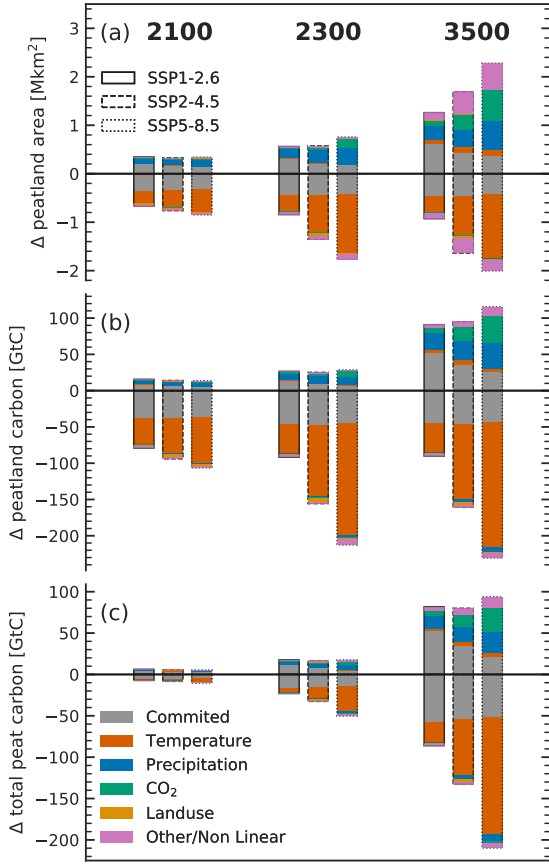

**Figure 10.** Mean driver contributions to gross positive and negative anomalies since 2014 in (a) global peatland area, (b) global peatland carbon, and (c) global total peat carbon as defined in section 2.2. Anomalies for all three future scenarios are with respect to 2014. Committed driving contributions can decrease, i.e., switching from driving to dampening, if the anomaly in the respective gridcell changes sign under the different scenarios

The late expansion after 2300 is driven by peatlands newly establishing in model gridcells with no previous peatland presence. In some gridcells in the northern high latitude and in south Asia, the historic climate change and atmospheric $CO_2$ rise
5   lead to the fulfilment of criteria for peatland establishment, targeting the peatland water and carbon balance. The stronger the boundary conditions change under additional future scenarios, the more gridcells, especially in the tropics, become able to support peatlands (see sect. 3.2.3). This change is driven by increases in precipitation, $CO_2$, temperature and the non-linear interactions between them. In the model, newly established peatlands start from a small seed and their growth is restricted





to 1% of their size per year. They thus reach noticeable size only centuries after their initial establishment. Outside of the model world, the speed of lateral expansion of growing peatlands is dependent on multiple factors including local topography, hydrology and peatland type (Charman, 2002; Ruppel et al., 2013). Topography can constrain lateral expansion velocities. Depending on terrain slopes, peat accumulation can be limited to a small area or depression for centuries to millennia until the

peat column grows tall enough or expand quickly over a flat plain (Bauer et al., 2003; Loisel et al., 2013; Broothaerts et al., 2014; Le Stum-Boivin et al., 2019). This heterogeneity and complexity in lateral expansion of newly established peatlands is not represented by the model used here. The magnitude and timing of the simulated late expansion should therefore be taken with care. However, the results suggest that historic and future climate change might create the potential for newly forming peatlands in regions where conditions have been mostly unsuitable before.

**3.4    Climate forcing uncertainty**

The spread between simulations forced with climate anomalies from the different CMIP6 climate models indicates a large climate anomaly related uncertainty in simulated peatland variables. This is in line with previous studies that also found a large uncertainty propagation from climate variables to peatland and carbon cycle variables in general (Stocker et al., 2013; Ahlström et al., 2017; Qiu et al., 2020; Müller and Joos, 2020).

The magnitude of uncertainties is regionally different with large uncertainties in the northern high latitudes for peatland area (Fig. S3) and total peat carbon (Fig. S4), and in the mid latitudes for peatland NEP (Fig. S5). The climate variables driving the uncertainty depend on the region and peatland variable in question. Linear regressions for each gridcell were used to investigate how the differences between the model climate anomalies translate to the simulated peatland variables. Differences in northern high latitude peatland area between simulations were found to be dominantly a factor of climate model temperature.

Warmer anomalies resulted in less peatland area in most gridcells, except for north east Asia, where warmer temperatures facilitate peat expansion in some gridcells (Fig. S3). In the tropics, the difference in precipitation is the best predictor for most gridcells, with anomalies from wetter models resulting in larger peatlands. Total peat carbon, determined by the balance between total accumulation and total decay of peat carbon, shows a similar regional pattern (Fig. S4). Northern high and mid latitude peat carbon is reduced with higher temperature anomalies as the area for accumulation declines and heterotrophic

respiration increases. In the tropics precipitation remains the dominant predictor, increasing the accumulation area and limiting respiration. For peatland NEP, precipitation minus evapotranspiration, as a measure of the moisture balance, resulted in a larger number of gridcells with significant (regression p-value < 0.05) results compared to temperature or precipitation alone (Fig. S5). Climate models resulting in a more positive water balance mostly also resulted in a higher peatland NEP due to the controls of peatland water table depth both on productivity and respiration. In permafrost regions also temperature on its own is a strong

positive factor for simulated peatland NEP.

The results show that the differences in peatland responses to different climate forcings can be explained mostly by the same drivers and mechanisms as the transient changes (discussed in sect. 3.3). They also reveal the large dependence of peatland and carbon cycle projections on key properties of climate models. Further constraining model climate sensitivity is thus essential to reduce uncertainty in carbon cycle projections. Here the climate model sample was selected to best represent the full CMIP6





ensemble and with this the fullest possible range of projections. However model performance, compared to different targets is highly variable (Harrison et al., 2014), and different sample selections or weighted ensemble medians might be preferable in future work, depending on the focus.

The simulations presented here are also subject to other large, but less quantifiable uncertainties. Structural and parameter
uncertainties, not only in the peat module, but through all components of the model, are unavoidable in simplified global models such as the LPX-Bern, especially on regional and local scales. Implementation of peatlands in DGVMs is still in its beginning and comprehensive model comparison and structural uncertainty evaluation is still mostly lacking. With the inclusion of peatlands into more and more DGVMs and earth system models, comparative studies might identify the most promising model developments and thus pave the way for more robust peatland and carbon cycle projections.

**4   Conclusion**

The dynamic global vegetation model LPX-Bern was used to estimate committed and projected mid to long term future changes of global peatland area and carbon under three different climate and land-use scenarios. A previously published transient simulation from the Last Glacial Maximum to the present (Müller and Joos, 2020) was used as the starting point for the future projections, accounting for the transient history and potential legacy effects of today's and former peatlands. LPX-Bern was
forced by climate anomalies from ten different CIMIP6 earth system models, selected to optimally represent the full CIMIP6 ensemble range. Peat carbon dynamics were analyzed for carbon in active peatlands (*peatland carbon*) and peat carbon in all land classes including former peatlands (*total peat carbon*), representing two land-atmosphere interaction bounding cases.

Averaged over 1995-2014 median global peatland area, peatland carbon and total peat carbon, are simulated to be 3.7 Mkm$^2$, 423 GtC, and 611 GtC respectively. This puts the modeled peatlands within the range of literature estimates, however with a
heavier weight on tropical peatlands than most estimates suggest. Simulations with constant 2014 boundary conditions revealed committed losses of northern peatland area (median: -8%) and peatland carbon (median: -10%) until the end of the century and beyond, with losses in Europe and eastern Canada partly compensated by peatland area expansion in eastern Asia, the western part of north America, and the tropics. These results suggest that past climate and land-use change has already led to regional changes in the environmental conditions that put a large part of today's northern peatlands at risk while potentially
improving conditions for others. With higher emission scenarios, global net losses in peatland area and carbon are increased with increases of losses in the northern latitudes and increasing gains in the tropics. Under the SSP1-2.6, SSP2-4.5, and SSP5-8.5 scenario northern peatland area is simulated to decrease by a median of -18 %, -41 %, and -61 % until 2300 respectively, with concomitant decreases of northern peatland carbon (-18 %, -47 %, and -65 %) and total peat carbon (-1 %, -3 %, and -7 %). These results illustrate the extent to which today's northern peatlands and their large carbon stocks are at risk from
future climate change. Estimated peat carbon loss here depends on the assumed emission bounding case and could be large in case the carbon is quickly released to the atmosphere (*peatland carbon*) or moderate to small if the carbon decays only slowly after ecosystem transformation (*total peat carbon*). To reduce this persisting uncertainty, additional research focus on the fate of carbon in former peatlands is needed, leading to dedicated model parametrizations for this carbon pool. In our





simulations, higher future emissions are clearly tied to a larger potential loss of peatland area and carbon, highlighting the role of fast emission reduction for peatland protection. While direct human disturbances of peatlands through drainage and land-use conversion can be partly mitigated by prompt peatland restoration and legal protection, indirect disturbances from anthropogenic climate change can only be limited by drastically cutting future emissions.

Beyond 2300 all simulations showed a delayed peatland expansion in grid-cells with no prior peatland presence. This delayed peatland expansion is especially pronounced in the tropics and the highest northern latitudes. Although the timing and magnitude of this expansion is most likely strongly model dependent, it illustrates the potential for new peat initiation in regions which where formerly unsuited for peat development. Towards the end of the simulations, medians for simulated global peatland area, peatland carbon and total peat carbon in the strong mitigation scenario exceed the ones of the 1500 and 2014

commitment simulations. The millennial scale potential for global peatland area and peat carbon storage is thus simulated to be larger under strongly mitigated climate change as under pre-industrial or present day conditions. This potential however is only realized after centuries to millennia of dominating negative effects, with large permanent losses of northern peatland area, peat carbon and ecosystem services provided by them. For the higher emission scenarios globally-aggregated negative changes in area and carbon persist until the end of the simulation.

Additional simulations with extended SSP scenario climate forcing from three different climate models showed that continuing transient forcing along a scenario trajectory can substantially change the simulated results. Extending the SSP1-2.6 scenario to 2300 with global temperature anomalies decreasing again after 2100 led to a reduction in the response, positive or negative, relative the standard scenario. The extension of SSP5-8.5 on the other hand lead to a drastically increased loss of peatland area and carbon due to the extreme increases in mean global temperature until 2300. These results highlight the

importance of extended emission pathways to project long term effects of anthropogenic climate change not only on peatlands but on the carbon cycle and the climate system as a whole. As the current century grows shorter the next phase of CMIP should aim to extend projections beyond the end of the century as a standard practice.

Driver contributions to future changes were analyzed using factorial simulations. Besides committed changes, increasing temperature was identified as the main driver of peatland area and carbon losses and increasing precipitation as the main driver

of gains. After 2300 influences of $CO_2$ and non-linear interactions on peat initiation become more apparent, when peatland area begins to expand more widely. Cloud cover was found to have only small influences on global peatland variables. Future changes in the net area under land use are small (< 11 %) in the scenarios compared to the historical changes and have a small impact on global peatlands in our simulations. Here a simplified assumption was taken with peatlands being affected by land-use change proportional to their size. However, this might not be the case if peatlands are directly targeted for conversion

to land-use areas. Future studies might try to integrate specific peatland - land-use conversion scenarios to better quantify the effect of potential future land-use conversion within a global modeling framework.

The spread between the simulations forced with different climate anomalies from the ten sample climate models, reveals that a large uncertainty is propagated from the climate anomalies to the global peatland and carbon cycle variables. Propagation was found to be mediated different regionally by temperature, precipitation or the combination of both. The uncertainty increases

with time even after climate forcing is kept constant due to the long response time scales important for peatlands. Even in





the case of the 2014 commitment simulations, which only see 40 years of slightly diverging climate anomalies, uncertainties grow large over time. This shows that small differences in climate forcing can propagate to large long term differences in peatland and carbon cycle variables. In future studies, uncertainties could be reduced by including a skill criterion into the climate model sample selection or the subsequent ensemble analysis. Structural model uncertainties are harder to quantify but

could potentially be equally large. A focus of future work must be to quantify these structural uncertainties in peatland model intercomparison projects and continue model development towards simple but robust formulations for dynamic peatlands on a global scale. Given the large and diverse uncertainties involved, the results presented here should be interpreted as a model analysis of potential risks, their transient evolution, environmental drivers and uncertainties, rather than as robust predictions.

    The climate and the terrestrial carbon cycle in the simulations presented in this study are uncoupled. The results however

suggest potentially large feedbacks between the simulated changes in global peatlands and the global carbon cycle and climate system. Carbon release to the atmosphere would additionally warm the climate leading to a positive feedback. Each 100 GtC released from peatlands would causes a warming of about $0.2^{o}$C (Allen et al., 2009). Other feedbacks not considered here include changes in the methane source, surface engergy balance, and surface albedo. Changing surface albedo and energy balance might lead to a warming given a transition from moss dominated boreal peatlands to dense forest (Helbig et al.,

2020a), however, strongly dependent on the actual vegetation succession. Decreasing methane emissions from reduced northern peatlands could be compensated or even superseded by increased emissions from expanding tropical peatlands. In addition, methane emissions increase with temperature (Turetsky et al., 2014). The sign of the methane feedback therefore is dependent on multiple factors. The absence of these potentially important feedbacks between peatlands and the climate system in the state-of-the-art future projections such as produced by the CMIP is a potential limit to formulating adequate climate policy. Future

work should focus on the production of fully coupled peatland-climate simulations to assess the magnitude of the potential feedbacks, as well as the integration of peatland modules into the next generation of earth system and integrated assessment models (Loisel et al., 2021).

    Taken together our study provides long term future projections of global peatland area and carbon, based on a transient spinup since the Last Glacial Maximum, and accompanied by an in-depth analysis of future scenarios, drivers and uncertainties. It

suggests that large parts of northern peatlands are at risk of both committed and future climate change and highlights the need for strong mitigation and protection efforts. The large uncertainties found call for continued model development and refinement. The long response timescales and potentially large climate feedbacks of peatlands stress the need for century-to-millennial scale coupled climate-peatland simulations.

**Appendix A: Additional Figures**

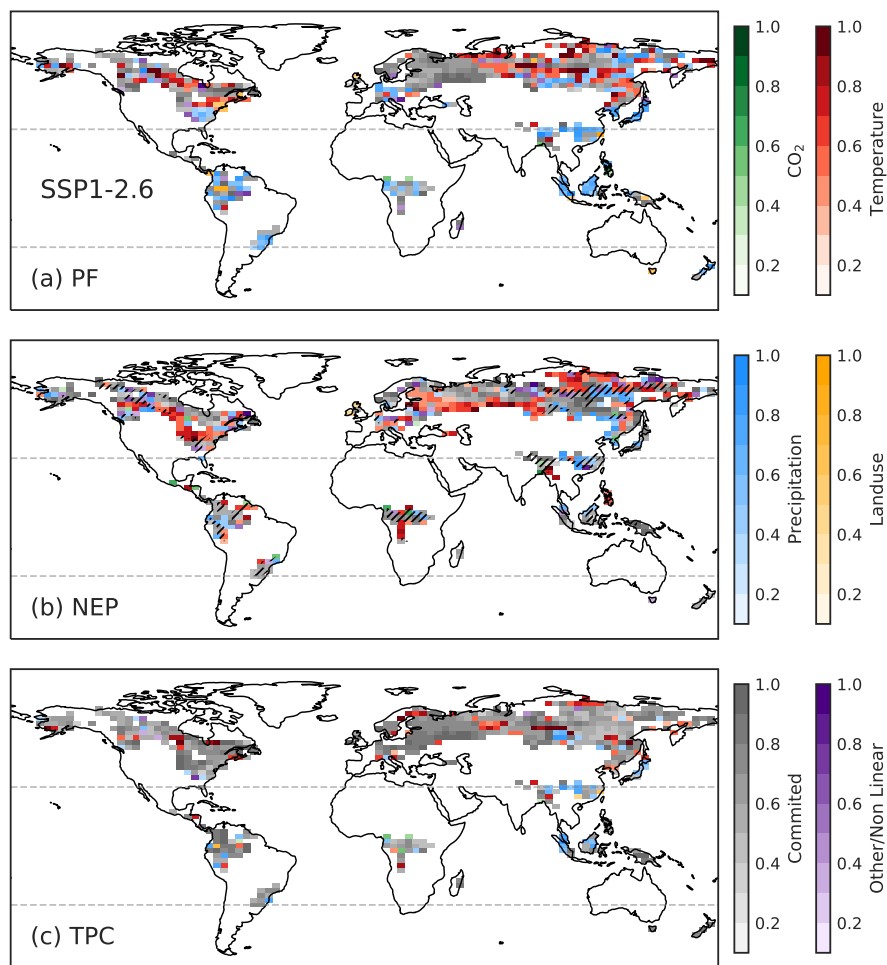

**Figure A1.** Dominant driver contributions to SSP1-2.6 anomalies at 2300 in (a) peatland area fraction (PF), (b) peatland NEP and (c) total peat carbon (TPC). Colors indicate the most important driver, and color shade the contribution of the respective driver on a scale from 0 (no contribution) to 1 (only contributor). Anomalies are calculated with respect to 1995-2014 averages



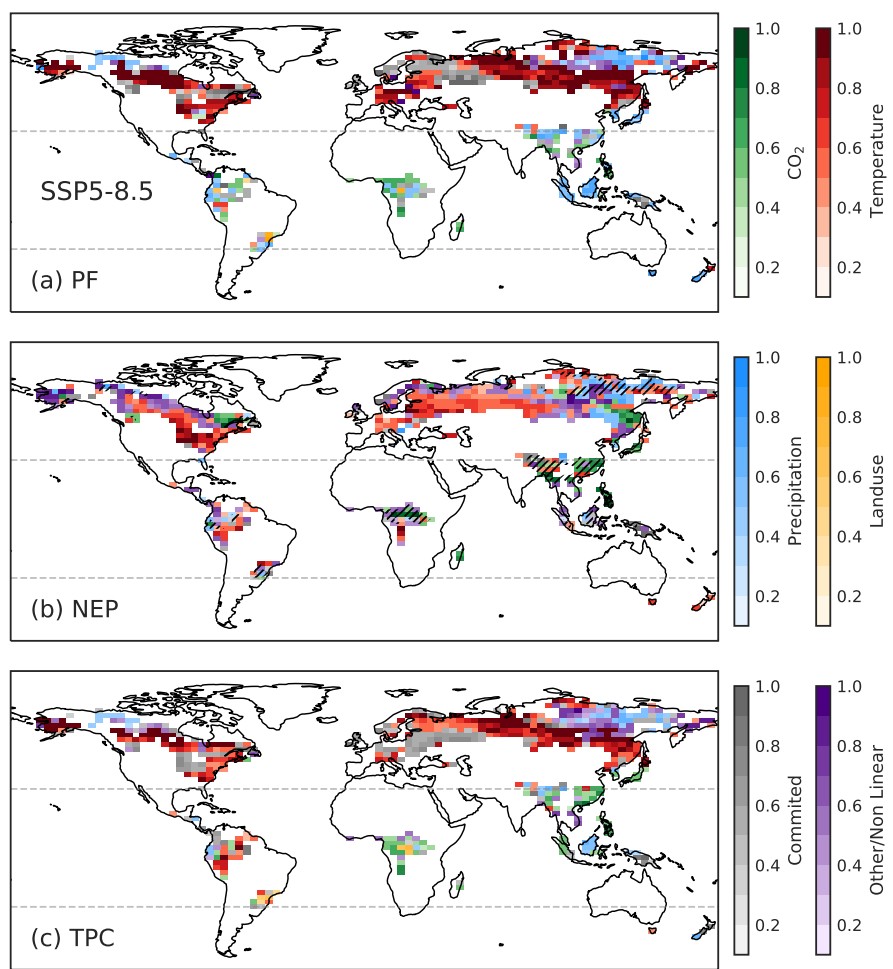

**Figure A2.** As Fig. A1 but for the SSP5-8.5 scenario





*Data availability.* Supplementary material with additional figures and tables are available online. LPX-Bern model output for variables and simulations presented here is available for download under https://doi.org/10.5281/zenodo.4627681.

*Author contributions.* JM and FJ designed the study. JM performed the simulations and the analyses in consultation with FJ. JM prepared the figures and wrote the paper with input from FJ.

*Competing interests.* The authors declare to have no competing interests

*Acknowledgements.* This project received funding from the Swiss National Science Foundation (no. 200020-172476) and from the European Union's Horizon 2020 research and innovation programme under grant agreement no. 820989 (project COMFORT, Our common future ocean in the Earth system – quantifying coupled cycles of carbon, oxygen, and nutrients for determining and achieving safe operating spaces with respect to tipping points) and under grant agreement no. 821003 (project 4C, Climate-Carbon Interactions in the Current Century). The work

reflects only the authors' view; the European Commission and their executive agency are not responsible for any use that may be made of the information the work contains. This study was undertaken as part of C-PEAT, a working group of the Past Global Changes (PAGES) project, which in turn received support from the Swiss Academy of Sciences and the Chinese Academy of Sciences. We thank Sebastian Lienert, Renato Spahni, and Benjamin Stocker for their contributions to the development of LPX-Bern. We acknowledge the World Climate Research Programme's Working Group on Coupled Modelling, which is responsible for CMIP, and the climate modelling groups for producing and

making available their model output.





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
