# Peer review of "Committed and projected future changes in global peatlands - continued transient model simulations since the Last Glacial Maximum"

_Biogeosciences, 2021_

## Author Comment (AC1)

We thank all of the reviewers for their very useful comments.

We provide a version of the manuscript showing the proposed changes and updated figures, including some additional minor wording changes and fixed typos, at the end of this document

**Typesetting:**
TEXT: Original reviewer comments
TEXT: Author response
*TEXT*: Changes in manuscript
Page and line numbers are given in respect to the original manuscript.

**Response to reviewer 1**

This study continues to extend a previous study by the same authors of simulating past global peatland dynamics into the future. Novel aspects of the study include the transient nature of the simulations from the previous LGM simulations, the long-term projections of the next few millennia, and the considerations of committed change from legacy changes. This is an important study that contributes to our understanding of long-term trajectories of large peatland carbon stocks.

The manuscript is well written and clearly organized. I have only some editorial suggestions that the authors could consider to improve the presentation.

We thank the reviewer for their positive words and constructive and helpful comments! We respond point by point below:

Specific comments:

Page 1:

Line 1: A possible different title could be:
"Committed and projected future changes in global peatlands—seamless transient model simulations since the Last Glacial Maximum"

These suggested changes because of the following considerations:
-"since" is more clear than "from", as "from" may need "to"
-"seamless projections from the LGM": not accurate, as it was not projections per se during most times of the last 21,000 years
-replace "seamless" with more conventional "transient" or "seamless transient"?
-"committed change" is a novel aspect of this study

-"future climate" is only parts of the drivers for change, so "future changes" is broader and more inclusive.

We agree with the argumentation and took the suggested title, with only a small change. The new title now reads:
*"Committed and projected future changes in global peatlands - continued transient model simulations since the Last Glacial Maximum"*

L5: change to "long-term projections". It is a good style to use hyphen between two nouns when they are used as adjective. For example, "long-term projections", "land-use conversion/change", "large-scale restoration", etc.

Changed throughout the text

L7: change to "three standard future scenarios", to distinct from "additional" scenarios later on?

Changed

L9-10: "Conditions for peatlands …. improve": the sentence is unclear. What and how?

The sentence was reworded and joined with the sentence before, to point out the effects on peatland area and carbon stocks. It now reads:
*"Large parts of today's active northern peatlands are at risk, whereas peatlands in the tropics and, in case of mitigation, eastern Asia and western North America can increase their area and carbon stocks."*

L10: change to "North America"

The writing of cardinal directions and regional names were corrected throught the manuscript: e.g. North America, northwestern Canada, Southeast Asia etc.

L12: these two additional CMIP6 scenarios are part of ten CMIP6 models? Perhaps better to distinct models, simulations and scenarios in the description. There may be the need for explanations of "three scenarios" (are they "standard scenarios"? in what way?) and "additional" "two CMIP6 scenarios", what the difference? Time durations? If so, spell out in the abstract as well.

The distinction between the different scenarios and simulation setups was indeed a bit confusing. To clarify the destinction between the standart and extended scenarios, as well as the extension until 7000 we extended the sentence on L8:

*"These projections are seamlessly continued from a transient simulation from the Last Glacial Maximum to the present to account for the full transient history and are continued beyond 2100 with constant boundary conditions."*
And reworded the sentence on L12:
*"Additional simulations forced with climate anomalies from a subset of climate models which follow the extended CMIP6 scenarios, transient until 2300, show qualitatively similar results to the standard scenarios, but highlight the importance of extended transient future scenarios for long-term carbon cycle projections."*
*We hope that the distinction is now clearer*

L15: change "peatland variables" to some other wordings, such as "peatland changes" or "peatland area and carbon stock"?

Changed to "*peatland changes*"

L20: change "accumulated organic carbon" to "accumulated organic matter". Using "matter" (materials) is better here.

Changed

Page 2:

L17: change "however" to ", however,". "however" often needs two commas when in the middle of a sentence, and when connecting two phrases, need ";" before it.

Changed throughout the text

L29: change to "southeast Asia"?

Changed to *"Southeast Asia"* throughout the text (see comment above)

Page 3:

L29 and L34: ", however,"

Changed

Page 4:

L2: change "over" to "during"?

Changed

L4: "total land area and land-use area"?

Changed

L9: use 21,000 years or 22,000 years for LGM, but to be consistent.

Changed to  22,000 consistent with the rest of the text.

L13: Maybe at the end of the Introduction, also describe the general approach of using different scenarios (standard vs. additional).

A sentence was added to explain the different scenarios and extensions beyond 2100. The last 3 sentences of the introduction now read:
*"Committed and future peatland responses to three different standard future emission and land-use scenarios are investigated using the DGVM LPX-Bern. Simulations are continued with constant forcing beyond 2100 to reveal delayed long-term effects on peatlands over the next five thousand years. Standard simulations are compared to additional simulations with extended transient scenario forcing until 2300 and constant forcing thereafter. Uncertainties and drivers are analyzed using multiple climate model forcings and factorial simulations."*

L17-18: after Joos (2020), change to ", which is described below briefly"

Changed to *"The model setup is mostly identical to Müller and Joos (2020), which is briefly described below."*

L20: change to "long-term carbon store"

Changed

Page 5:

L5: "due to fluctuations/changes in peatland area"?

Changed to "*fluctuations in peatland area*"

L15: change "like e.g." to "such as"

Changed

L20: change "like" to "such as"

Changed

Page 6:

L1: ", however,"

Changed

L9-10: "of actively growing peatlands"

Changed to "actively accumulating" instead, to avoid association with area growth

L10-11: change to "… all carbon that have accumulated, including by former peatlands and peatlands transformed to land-use area"

We did not simplify the sentence as much as suggested, because we feel the information that only catotelm carbon is tracked is important. However, we tried to reword the sentence and hope it lead to increased clarity. The sentence now reads: *"2) total peat carbon, which is calculated in post processing and represents all carbon in the catotelm of active peatlands and organic, not yet decomposed carbon that was at some point sequestered into a catotelm on peatlands transformed to land-use areas and other former peatlands."*

L22: be consistent with the age of the LGM: 22,000 or 21,000 years. See comment above.

See response to comment above

L30: change to "a transient land-use history", otherwise "land use" (no -) if not used to modify another noun

Changed to "land use". Also changed throughout the text where applicable

L34: change to "each model of a ten member climate model ensemble"? "member" is unclear here. There are other cases later on.

Changed (see also comment below)

Page 7:

Figure 1 caption: change "full arrows" to "solid arrow"? Perhaps better to describe as light color rather than "transparent" (as no overlay, transparent is not apparent)

Changed

Page 8:

L17: change "sub-sample" to "subset". Throughout the manuscript, perhaps you need to be clear about how to describe the ensemble (22 models in CMIP6), 10 models from this ensemble, and individual model (maybe just call it model, rather than "sample").

The naming was indeed confusing. Throughout the manuscript the term "sample" was replaced with "ensemble" referring to the 10 climate models used in this study. Only in section 2.5 and when it is compared directly to the "full CMIP6 ensemble" (22 models), it is referred to as "ensemble subset" or just "subset". Furthermore, the models providing the extended scenarios have no specific naming anymore and are just referred to as: "The models that provided the extended scenario output [...]" (P21L33)
We hope the changes lead to simpler and clearer naming conventions.

L20: delete "searched and"

Changed

L22 and L24: "included in a sample": ensemble? Group?

Changed "sample" to "subset" (see also comment above)

Page 9:

Table 1: "Date DOI": should provide doi instead references?

The DOI are part of the reference in the bibliography. Changed to "*Data reference*"

L16: change "mayor" to "major"

Changed throughout the text

Page 11:

L4: change "land-use and land-use change" to "land use and land-use change"

Changed

Page 12:

Headings 3.1 and 3.1.1: change "Historic" to "Historical"

Changed

Page 13:

L11-12: Northeastern Canada, East Asia, northwestern Canada, northeastern Asia, Southeast Asia??

Changed throughout the text (see comment above)

L25: change "historical" to "historic"

Changed from "historic" to "historical"

**Response to reviewer 2**

In their manuscript – a follow-on study to their previous publication covering the time from the LGM to the present – Müller & Joos investigate the future evolution of peatlands in the LPX-Bern DGVM. They show and analyse the development of peatlands under a spread of climate forcings from an ensemble of CMIP6 models.

Overall, this is a superb manuscript analysing the future of global peatlands. There are, of course, a few minor issues to resolve, but the manuscript is nearly ready for publication.

We thank the reviewer for his positive words and constructive and helpful comments! A point by point response follows below:

Initially I had strong reservations about the manuscript, as the authors only used climate model forcings for the years 1975-2100 in most cases, and 1975-2300 in three cases, while they analysed the peatland development for the next 5000 years. This is not the authors' fault, as the output from most GCMs in CMIP6 is only available until 2100, very few modelling groups provide extended scenarios until 2300, and none provide output beyond 2300.

Potentially, the repeated climate forcing from the last years data were available might bias results quite strongly. However, experiments we had performed with MPI-ESM for the next millennium confirm that the bias introduced this ways is rather small, as climate remains quite stable after 2300 in most scenarios. Nonetheless, I suggest that the authors elaborate some more on the potential shortcomings of their use of repeated climate anomalies for several millennia.

We are aware that constant forcing for millennia is unrealistic. However, despite the absence of transient forcing, we wanted to show and investigate the delayed long-term response of peatlands after reaching a certain climate state. The results for the constant climate extensions thus should not be seen as predictions, but rather as a sensitivity study of committed changes, similar to the

simulation with constant 2014 climate. We changed and expanded the text in several places in the manuscript to underline this point:

Section 1 P4L12:
A sentence was added to explain the different scenarios and extensions beyond 2100. The last 3 sentences of the introduction now read:
*"Committed and future peatland responses to three different standard future emission and land-use scenarios are investigated using the DGVM LPX-Bern. Simulations are continued with constant forcing beyond 2100 to reveal delayed long-term effects on peatlands. Standard simulations are compared to additional simulations with extended transient scenario forcing until 2300. Uncertainties and drivers are analyzed using multiple climate model forcings and factorial simulations."*

Section 2.3 P6L3:
The description of the standard simulation setup was slightly changed. The paragraph now reads:
*"Simulations corresponding to three different CMIP6 scenarios start from the year 2015. One strong mitigation (SSP1-2.6), one middle of the road (SSP2-4.5) and one high emission scenario (SSP5-8.5) were selected to represent the scenario range. The standard CMIP6 scenarios end in the year 2100. To investigate the delayed long-term responses of peatlands the forcing is extended into the future with a detrended version of the last 30 years of each time series repeated over almost 5.000 years until the year 7000."*

Section 2.3 P7L10:
The description of the extended simulation setup was slightly changed. The paragraph now reads:
*"CMIP6 also includes extended versions of the scenarios SSP1-2.6 and SSP5-8.5 that range until 2300. At the time of this study, however, only three climate models had provided output for these extended scenarios. Climate projections of these three models alone are not representative of the full CMIP6 scenario. They were, however, included in the ten-member climate model ensemble used here (see sect. 2.5 and Fig. S1) and additional simulations with transient climate and CO2 forcing until 2300 were performed to compare results to the standard simulations."*

Section 3.2.2 P17L13:
A half sentence was added:
*"Taken together these results suggest a likely net loss of global peatland area as well as carbon until the end of the century, driven mostly by northern peatlands, even under the strongest mitigation scenario and continued net loss up to 2300 for the scenarios SSP2-4.5 and SSP5-8.5**, with climate assumed to remain constant after 2100.***"

Section 3.2.3. P20L17:

A sentence was added at the beginning:
"*The continuation of the simulations for several millennia under constant boundary conditions, reveals the delayed long-term responses of peatlands to the previous changes in forcing.*"

Section 3.2.4 P21L6:
The sentence was reworded and split in two:
"*The assumption of stable climate and atmospheric CO2 levels over millennia after 2100 is a highly idealized one and not suited for predictions. The extended simulations are rather intended to reveal delayed responses and long-lasting effects in the slow-reacting peatland system.*"

Section 3.4 P25L4:
A sentence was added to the paragraph about additional uncertainties:
"*Keeping the forcing constant after 2100 or 2300 is an idealization with true climate dynamics depending on highly uncertain factors such as future social and economic dynamics and potential tipping points.*"

Section 4 P26L27:
A half sentence was added:
"*Under the SSP1-2.6, SSP2-4.5, and SSP5-8.5 scenario, **assuming constant climate, CO2, and land-use forcing after 2100,** northern peatland area is simulated to [...]*"

Section 4 P27L5:
The first sentence of the paragraph was changed:
"*All simulations showed delayed peatland responses beyond 2300 under constant climate and environmental forcing, most notably a delayed peatland expansion in grid-cells with no prior peatland presence.*"

I'd also suggest that the authors put more of a focus on the extended scenario experiments, as the climate forcing changes quite drastically between 2100 and 2300, certainly for the high radiative forcing scenario.

As the three climate models that provided transient output until 2300 are not representative of the whole CMIP6 ensemble and as there is no extended version of the, maybe most realistic middle of the road scenario SSP2-4.5, we chose to focus on the simulations with transient climate forcing until 2100 and use the simulations with extended transient forcing until 2300, as a showcase of how results could change given continued transient forcing.
We changed and expanded the text in two places in the manuscript to underline this point:

Section 3.2.4 P22L1:

A discussion of expected ensemble changes under transient forcing until 2300 was added to the section summary:

*"In case the discussed differences in the peatland response to constant and transient forcing after 2100 are similar for all climate models, assumptions can be made about how the ensemble results would change in the case of transient climate until 2300. Transiently extending SSP1-2.6 would likely lead to an overall weaker peatland response, leading to a reduced model spread and less peat carbon loss in the ensemble median. For the SSP5-8.5 we would expect a larger loss of peatland area and carbon over the whole ensemble, shifting medians to larger negative anomalies."*

Section 4 P27L19:
A sentence was added in the Conclusion:
*" Assuming similar results when extended scenarios would be used for the whole ensemble, model spread and median peat carbon loss are expected to be smaller under SSP1-2.6, and median loss of peatland area and carbon is expected to be substantially larger under SSP5-8.5."*

This was the one (semi-) major issue I have with the manuscript. In addition, I was surprised that the authors deal somewhat half-heartedly with one of the most exciting features of their results: The expansion of peatlands after 2700. Yes, it is clear that the criteria for establishment of peatlands must be fulfilled (page 24) – that is implicit in the design of the model – but which ones? What exactly leads to the establishment / growth? Precip increase? Permafrost thaw? NEP increase? If it is too complicated a picture, would a map be possible?

It is indeed an interesting question, what exactly leads to the initiation of new peatlands in the model. In a given gridcell, and at a certain time, multiple initiation criteria could simultaneously hinder initiation. Furthermore, changes in multiple environmental drivers could lead to the later fulfillment of single initiation criteria or changes in a single environmental driver could lead to the fulfillment of multiple criteria. To simplify the picture, rather than focusing on the specific criteria which hindered earlier peat initiation, we chose to visualize the dominant environmental drivers, that lead to peat establishment in the respective gridcells. We added the supplementary figure A3, in which these dominant drivers are shown. Furthermore, we updated the paragraph starting at P24L3 to include information from figure A3. It now reads:
*"The late expansion after 2300 is driven by peatlands newly establishing in model gridcells with no previous peatland presence. In some gridcells, the historical and future climate change and atmospheric CO2 rise lead to the fulfillment of criteria for peatland establishment, targeting the peatland water and carbon balance. The stronger the boundary conditions change under future scenarios, the more gridcells, especially in the tropics, become able to support peatlands (see sect. 3.2.3). Initiation of new peatlands in the tropics is mostly driven by CO2 fertilization nudging the carbon balance over the initiation thresholds (Fig. A3). In northeastern Asia, the temperature rise is the most*

*prominent initiation driver, whereas in northwestern Canada mostly precipitation increases drive the initiation of new peatlands.*"

Finally, there are some minor wording suggestions:

- please do a global search and replace changing all instances of "mayor" (German: Bürgermeister) to "major" (German: größer)

Changed throughout the text

- the same goes for historic: Please change to historical

Changed throughout the text

- the authors sometimes mention the "sample members", for example page 7, line 16, page 8, line 3: I'd suggest to use "ensemble member", as this is the genrally accepted usage.

The naming was indeed confusing. Throughout the manuscript the term "sample" was replaced with "ensemble" referring to the 10 climate models used in this study. Only in section 2.5 and when it is compared directly to the "full CMIP6 ensemble" (22 models), it is referred to as "ensemble subset" or just "subset". Furthermore, the models providing the extended scenarios have no specific naming anymore and are just referred to as: "The models that provided the extended scenario output [...]" (P21L33)
We hope the changes lead to simpler and clearer naming conventions.

- sometimes "at the year XXXX" is used – usually one uses "in the year XXXX"

Changed throughout the text

- page 2, line 1: remove first instance of "global"

Changed

- page 2, lines 19 and 32: past tense of "lead" is "led" (I think this also occurred in a few other places)

Changed

- page 17, line 24: "main reasons for…" (reason plural, not singular)

Changed

- page 17, line 29: particularly strong

Changed

- page 20, line 10: latitude instead of latitudinal

Changed

- page 23, line 2: "depending on multiple…"

Changed

- page 23, line 13: "area than would be" (than instead of as)

Changed

- page 23, line 23: "as well as in..."

Changed

- page 23, line 33: "scenarios than for the historical…" (than instead of as)

Changed

- page 26, line 15: CMIP6, not CIMIP6

Changed

- page 27, line 11: "climate change than under pre-industrial" (instead of as)

Changed

- page 27, line 34: "mediated different regionally"??? I don't understand this.

The sentence was reworded for greater clarity and now reads:
*"Depending on the region, uncertainty was propagated mostly by temperature, precipitation, or a combination of both."*

- page 28, line 12: "cause", not "causes"

Changed

[revised manuscript text omitted]